# Somatodendritic consistency check for temporal feature segmentation

Toshitake Asabuki[1] & Tomoki Fukai[1,2,3 ✉]

The brain identifies potentially salient features within continuous information streams to process hierarchical temporal events. This requires the compression of information streams, for which effective computational principles are yet to be explored. Backpropagating action potentials can induce synaptic plasticity in the dendrites of cortical pyramidal neurons. By analogy with this effect, we model a self-supervising process that increases the similarity between dendritic and somatic activities where the somatic activity is normalized by a running average. We further show that a family of networks composed of the two-compartment neurons performs a surprisingly wide variety of complex unsupervised learning tasks, including chunking of temporal sequences and the source separation of mixed correlated signals. Common methods applicable to these temporal feature analyses were previously unknown. Our results suggest the powerful ability of neural networks with dendrites to analyze temporal features. This simple neuron model may also be potentially useful in neural engineering applications.

[1] Department of Complexity Science and Engineering, University of Tokyo, Kashiwa, Chiba 277-8561, Japan. [2] Okinawa Institute of Science and Technology, Onna-son, Kunigami-gun, Okinawa 904-0495, Japan. [3] RIKEN Center for Brain Science, Wako, Saitama 351-0198, Japan. ✉email: tomoki.fukai@oist.jp

C ognitive functions of the brain entail modeling of externally or internally driven dynamical processes. For this modeling, the brain has to identify the salient temporal features of continuous information streams. How the brain conducts this time-series analysis remains unknown, but the component processes necessary for the analysis are partly known. The process by which frequently recurring segments of temporal sequences are concatenated into single units that are easy to process is called chunking or bracketing[1]. Chunking underlies sensory scene analyses, motor learning, episodic memory, and language processing[2–6]. In predictive coding[7–9], the brain may chunk information in bottom-up and top-down pathways to identify variables relevant to the hierarchical Bayesian modeling of mental processes. Another important class of temporal feature analysis is blind source separation (BSS: related to the so-called cocktail party effect) in which the brain separates mixed sensory signals (typically auditory) from multiple sources in order to recognize the individual sources[10]. Despite their functional importance, the mechanisms by which neural circuits in the brain analyze and learn temporal features remain largely unclear. Whether different temporal feature analyses require specialized network architectures and learning rules is also unknown.

In this study, we introduce a novel solution to these fundamental problems of brain computing. We show, in a two-compartment neuron model, that the minimization of information loss between dendritic synaptic input and a neuron's own output spike trains enables efficient learning of clustered temporal events in a completely unsupervised manner. This learning proceeds intracellularly and can be viewed as a self-supervising process in which a single neuron (more precisely, the soma) generates an appropriate supervision signal to learn the spatiotemporal firing patterns repeated in upstream neurons (projecting to the dendrites of the neuron). The resultant learning rule conceptually resembles Hebbian learning with backpropagating action potentials, which experimental results[11–15] have demonstrated to be crucial to synaptic plasticity in cortical neurons. Importantly, our learning rule exploits the fact that neuronal adaptation is able to maintain somatic membrane potential in a regime where spiking has high information content[16–19]. Therefore, the gain and threshold of the somatic transfer function in our model are adapted in a history-dependent manner.

To our surprise, a family of competitive networks of the proposed neuron model can perform a variety of unsupervised learning tasks ranging from chunking to BSS, which were previously performed by specialized, distinct networks and learning rules. Members of this family have the same network architecture but different network parameters (e.g., synaptic weights). We emphasize that some chunking tasks solvable with our model (and also by humans) are difficult for conventional machine learning methods due to uniform transition probabilities between consecutive items[5]. Furthermore, the same network model successfully separates the mixed signals of highly correlated sources, namely musical instruments playing the same note. BSS has been extensively studied in machine learning[20–23], but how the brain solves this problem is not fully understood. Our results provide suggestions for computational principles which may underlie the wide range of subconscious temporal feature analyses by cortical networks and the active role of dendrites in these processes.

Our algorithm builds on ideas introduced by the two-compartment learning rule of Urbanczik and Senn[24], expanding the scope of neural computing towards slow-feature analysis (SFA[25]) and independent-component analysis (ICA) based on temporal correlations[26]. A central feature of our learning rule is that synaptic weights on the dendrite are changed such that the somatic membrane potential fluctuates with unit variance around a target value. Our formulation is inspired by the observations

that neuronal adaptation shifts the neuron always toward a regime of efficient information transmission[16–19].

## Results

**The minimization of regularized information loss.** Our model learns temporal features of an input based on a novel learning rule which we call minimization of regularized information loss (MRIL). Suppose the dendrite attempts to predict the responses of soma. In short, MRIL achieves this by minimizing the information loss (within a certain recent period) when the somatic activity is replaced with its model, that is, the dendritic activity driven by given synaptic inputs, the loss can be easily minimized if the somatic responses are well predicted. This will be the case when the neuron learns to selectively respond to temporal patterns recurring in synaptic input. Figure 1a schematically illustrates the present learning rule in a two-compartment spiking neuron model. Mathematically, MRIL minimizes the Kullback–Leibler (KL) divergence between the probability distributions of somatic and dendritic activities (see Methods for mathematical details). Note that in the resultant learning rule the somatic response is fed back to the dendrite to train dendritic synapses. These processes may be regarded as a consistency check between the soma and dendrite. Although the underlying biological mechanisms are not modeled here, backpropagating action potentials may provide such a feedback signal in cortical pyramidal neurons[27].

The division of labor between the soma and dendrite was previously modeled with a teaching signal given explicitly or implicitly to the soma[24]. Unlike the previous model, our model modulates the gain and threshold of somatic responses according to the recent history of somatic responses. These modulations enable the model to avoid a trivial solution to the learning rule (see Methods), and therefore ensure successful learning of nontrivial temporal features. Differences between the present and previous models will be further discussed later.

Our learning rule (Eq. 16 in Methods) looks similar to maximum likelihood estimation[28], a well-studied framework of supervised learning. However, there is a conceptual difference between them. In maximum likelihood estimation, the target data distribution (somatic activity) is provided externally as teaching signals. By contrast, our model simultaneously learns the probability distributions of input and output data without teaching signals. The consistency between the two data sets constrains the self-supervised learning, thereby avoiding an overly redundant or an overly simplistic categorization of temporal inputs. We emphasize that MRIL fits particularly well with neurons with dendrites, but the principle is generic and applicable to a broad range of information processing systems.

**Learning patterned temporal inputs in single neurons.** We first demonstrate that the two-compartment neuron model detects the salient temporal features recurring in synaptic input. Learning to detect and discriminate repeated temporal input patterns is crucial for various cognitive functions such as language acquisition[29,30] and motor sequence learning[2–4,31]. In Fig. 1b, presynaptic spike trains intermittently repeated three fixed temporal patterns of 50 ms each with equal probabilities of occurrence. These patterns may be regarded as chunks. As learning of the temporal input proceeds through the consistency check between the soma and dendrite, a single neuron gradually learned to respond selectively to an input pattern (Fig. 1c, d). The neuron learned one of the input patterns with approximately equal probabilities among the trials, although it responded to more than one input pattern in some trials (Fig. 1e). We note that all presynaptic neurons had the same average firing rates, which were

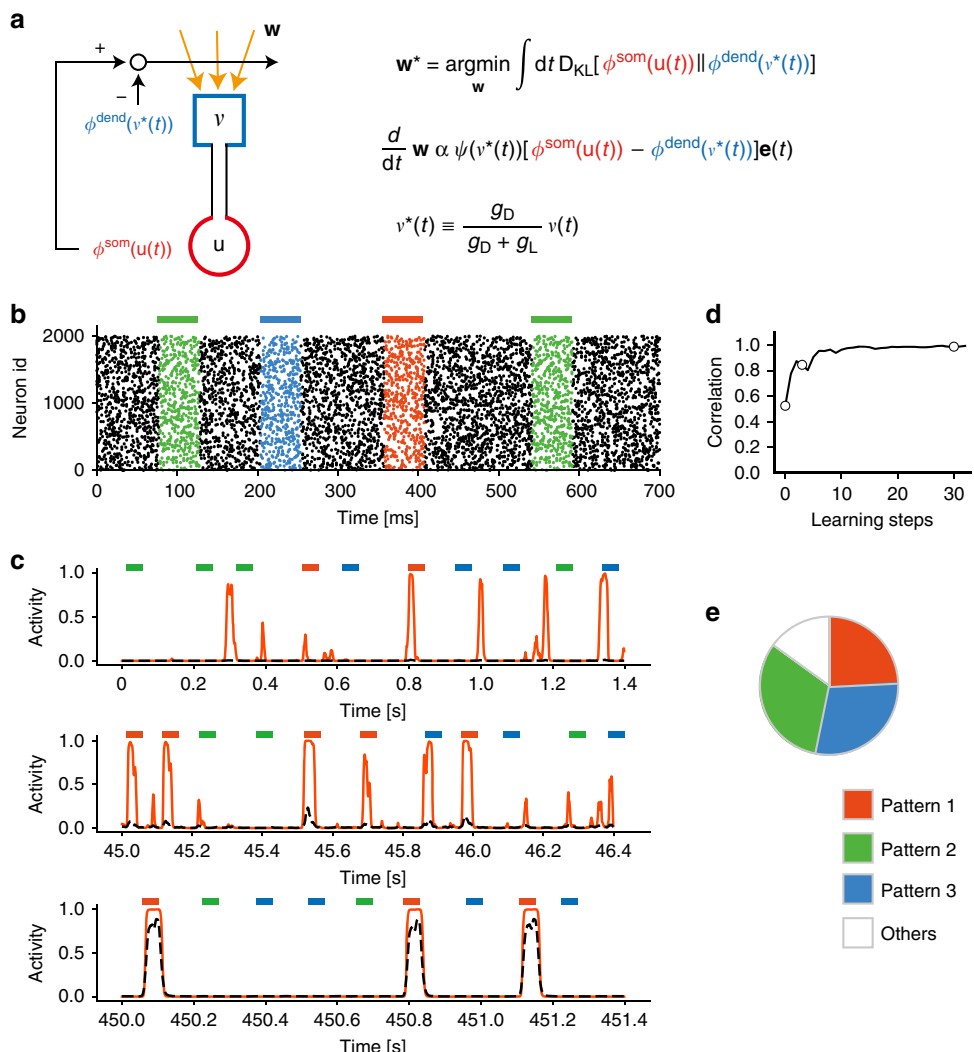

**Fig. 1 Unsupervised learning in two-compartment neurons. a** The model neuron consists of somatic and dendritic compartments and undergoes MRIL learning. The dendritic component receives Poisson spike trains, and the somatic membrane potential is given as an attenuated version of the dendritic membrane potential. Output of the soma backpropagates to dendritic synapses as a self-teaching signal. Learning stops when the dendrite minimizes the error between its prediction and the actual somatic firing rate. **b** Three frozen spatiotemporal patterns (red, blue, and green) were repeated in irregular spike trains from 2000 input neurons. **c** A two-compartment neuron selectively learned one of the recurring patterns. Examples of the somatic (red) and dendritic (black dashed lines) activities are shown at the initial (top), middle (middle), and final (bottom) stages of learning. **d** Learning curve is shown, with circles indicating the time points at which the examples were drawn. Instantaneous correlations were calculated between the activities of the dendrite and soma every 15 s during learning. **e** The fraction of trials in which a single neuron model learned a selective response to one of the three repeated spike patterns is shown. The number of trials was 100. In some trials (Others), the neuron had more than one preferred pattern, i.e., the peak response to the second preferred pattern was greater than 50% of that to the most preferred pattern.

constant during the entire task period (Methods). Therefore, the discrimination does not rely on differences in firing rates. Cortical neurons are actually capable of discriminating temporal inputs and generating sequence-selective spike outputs, although the synaptic sequences tested in the experiment were relatively simple[32].

**Automatic chunking with MRIL and inhibitory STDP.** Next, we considered a competitive network of the two-compartment model neurons receiving similar presynaptic spike trains (Fig. 2a). To study whether chunk-specific cell assemblies can be formed, we made recurrent inhibitory connections among these neurons modifiable by inhibitory spike timing-dependent plasticity (iSTDP; Fig. 2b). For near synchronous presynaptic and postsynaptic spikes, changes in inhibitory weights are negative in our

iSTDP rule. Consequently, this rule weakens inhibition between two neurons when both of them respond to the same temporal feature, as shown below. The use of this plasticity rule for lateral inhibition is realistic given this type of STDP has been found at cortical excitatory synapses on inhibitory interneurons[33] and at inhibitory synapses in the hippocampus[34]. In either case, inhibitory circuits will exhibit the desired changes. Note that inhibitory weights were restricted in the positive regime (Methods). During learning, each neuron gradually increased coherence between the somatic and dendritic activities (Fig. 2c). The postsynaptic neurons self-organized into three neuron ensembles, each detecting one of the input activity patterns (Fig. 2d), through iSTDP which enabled mutual inhibition between the neural ensembles (Fig. 2e). The strength of lateral inhibition needs to be within an appropriate range, as too strong (Supplementary Fig. 1a) or too weak (Supplementary Fig. 1b) inhibition failed to

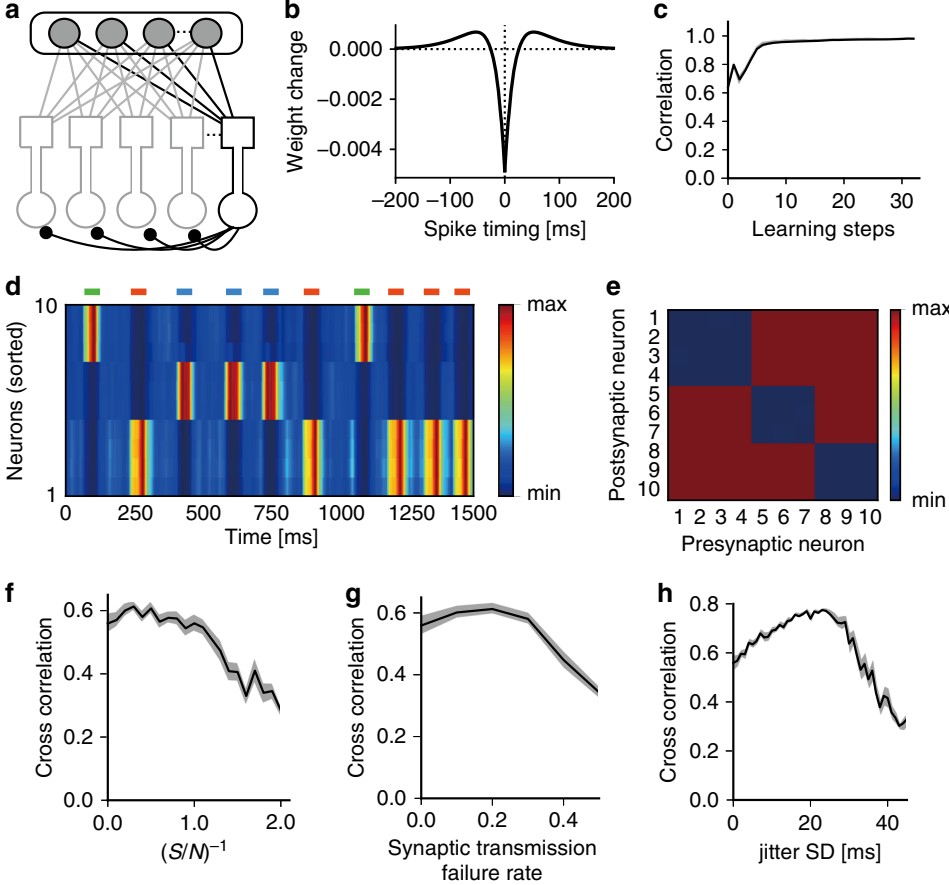

**Fig. 2 Formation of temporal feature-specific cell assemblies. a** A competitive network of two-compartment neurons was used throughout this study. The input layer consists of Poisson spiking neurons and the output layer is comprised of the two-compartment neuron models. In this particular example, input neurons received presynaptic spikes trains similar to those shown in Fig. 1b. **b** Window function of the iSTDP implemented at lateral inhibitory connections. Here spike timing refers to the time advance of postsynaptic firing from the preceding presynaptic input. **c** The average correlation between the somatic and dendritic activities is plotted against learning step. Shaded area represents the s.d. **d** Phasic responses of output neurons are shown. Horizontal bars show the intervals in which three chunks (green, red, and blue) were presented. The responses are sorted according to the neurons' onset response times and indicate the emergence of chunk-specific cell assemblies. **e** Post-learning synaptic weight matrix of lateral inhibition. The correlations between reference responses and actual output responses were evaluated in the presence of **f** contamination by background presynaptic spikes, **g** failure in synaptic transmissions, and **h** timing jitters in the target spiking patterns. Each reference response takes unity during the presentation of the corresponding chunk and zero otherwise. The ordinates refer to the inverse of the number ratio of background spikes to target-specific spikes in **f** and the s.d. of spike timing jitters in **h**. The mean (thick line) and s.d. (shaded area) over 20 trials are shown. The correlations are shown for the maximally correlated pairs of cell assemblies and chunks (i.e., preferred chunks).

generate chunk-specific cell assemblies. The regularization parameter $\gamma$ (see Methods) also has to be in an appropriate range, as values which were too large suppressed all neural responses and those which were too small did not generate selective responses to chunks (Supplementary Fig. 1c).

Weights of mutual inhibition were strengthened rather than weakened when a neuron pair fired synchronously in several previous models[35,36]. We therefore tested whether and how the conventional iSTDP rule works in the above chunk-detection task (Supplementary Fig. 2a). The conventional iSTDP rule generated variety of complex response patterns (Supplementary Fig. 2b). A small portion of neurons expressed chunk-specific responses (e.g., neurons 3, 4 and 8). However, some neurons responded to more than one chunk (e.g., neurons 1 and 10) and other neurons to chunks and random inputs almost arbitrarily (e.g., neuron 5). Inhibitory weight matrix also showed no obvious cell-assembly structure (Supplementary Fig. 2c). Therefore, the iSTDP rule shown in Fig. 2a is thought to be more suitable than the conventional one for the present chunk-detection task.

The ability of the network model to learn recurring input patterns was assessed with various types of biological noise. Background presynaptic spikes degraded the performance as the signal-to-noise ratio decreased (Fig. 2f), whereas learning was optimal at finite noise levels with synaptic transmission failure (Fig. 2g) and with jitters in presynaptic spike timing (Fig. 2h). We speculate that this disparity may reflect the different underlying noise structures. Background spikes were uncorrelated with the recurring input patterns and merely contaminated the signals, whereas transmission failures and timing jitters yielded noise patterns which were correlated with the input and thus enhanced the sampling during training. Therefore, the two types of noise are thought to induce data augmentation. Presynaptic noise may also induce a regularization effect during learning[37]. However, this effect was unlikely to be prominent in our model as not all types of presynaptic noise improved the learning.

The above results may account for the perceptual ability of humans to detect the recurrence of frozen noise patterns embedded in a noisy auditory signal[38]. As in Fig. 1b, both repeated and

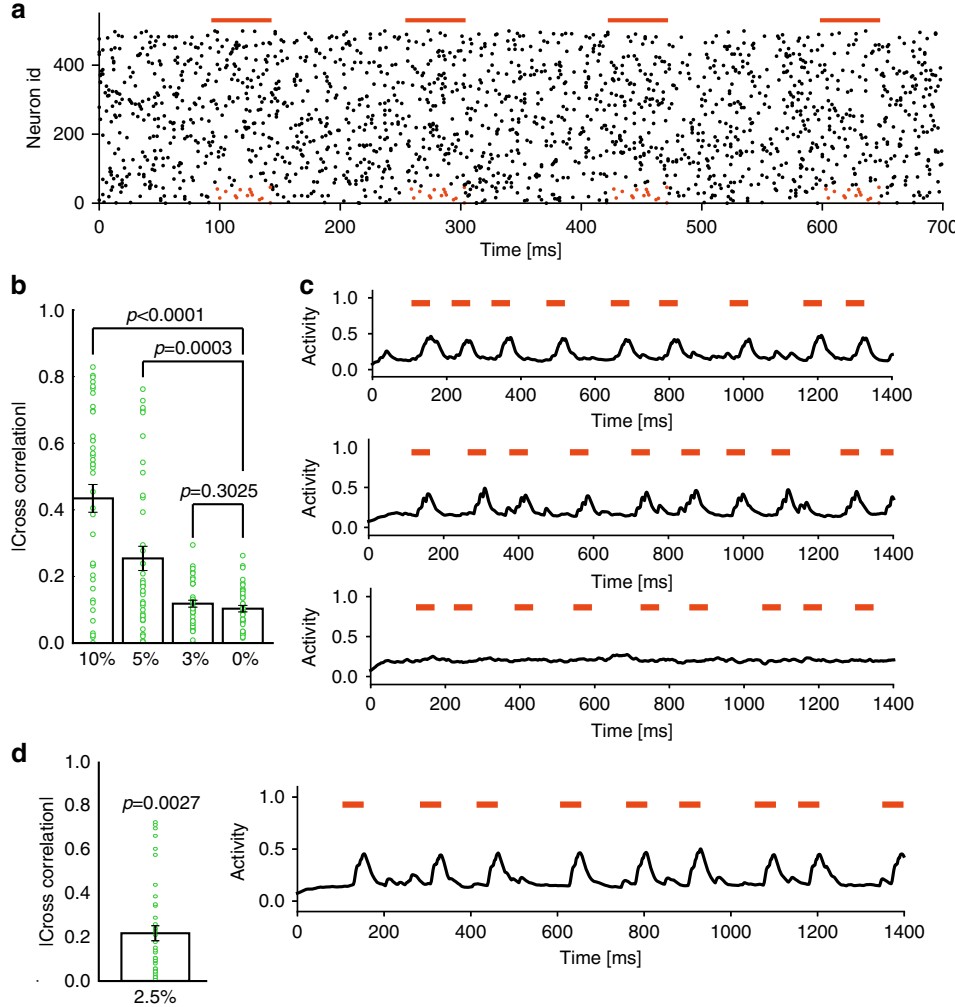

**Fig. 3 Detection of cell assembly patterns from neural population data. a** Spike sequences with a fixed spatiotemporal pattern (red) of 50 (10%) neurons were repeatedly activated (red horizontal bars) by Poisson spike trains of 500 presynaptic neurons. Other cases with 25 (5%) and 15 (3%) such neurons were also examined. **b** Average correlations over 40 independent trials are shown between chunk-selective responses and the corresponding reference patterns. Vertical bars are standard errors and green circles show data points. *P*-values were calculated by two-sided Welch's *t*-test. **c** Examples of chunk-selective neuronal responses in the 10% (top), 5% (middle), and 3% (bottom) cases. **d** A recurring firing pattern of 25 neurons was embedded into input spike trains of 1000 presynaptic neurons. The average and standard error of the input-output correlation with overlaid data points (left) and a typical response after learning (right) are shown. *P*-value (two-sided Welch's t-test) indicates a significant difference from the 0% case in **b**. As in **b**, 40 independent simulations were performed.

background auditory signals may be represented by irregular synaptic inputs to the auditory cortex. However, the subjects from this report[38] learned the noise without extensive training, indicating that the learning mechanisms might differ from the method presented here.

We may use the present network model in analyzing large-scale neural activity data. To show this, we performed similar simulations using synthetic data in which only a small fraction of presynaptic neurons (from a total 500) constituted a recurring pattern (Fig. 3a). (We note it is unlikely a large portion of recorded neurons participate in recurring cell assemblies in real data.) Learning was successful when the fraction of presynaptic neurons constituting the recurring pattern was 10% or 5%, but unsuccessful at 3% (Fig. 3b, c). We then considered the case where the total number of presynaptic neurons was 1,000 and 25 neurons (2.5% of all neurons) belonged to a patterned activity. Interestingly, the network still succeeded to learn the pattern, indicating that successful learning requires a minimal absolute number, but not a minimal fraction, of pattern-encoding presynaptic neurons (Fig. 3d).

Previously, STDP was used to detect repeated spike sequences[39,40]. We compared the detection performance between the present model and a STDP-based model[39] (see Supplementary Fig. 3a, b). Both models exhibited high success rates when recurring cell assemblies made up a large portion of presynaptic neurons. An interesting difference was found when only a small portion of presynaptic neurons participated in the cell assemblies. In such cases, our model outperformed the previous model (Supplementary Fig. 3c).

We further examined the ability of our network model in learning a variety of information streams. First, we applied random sequences of three chunks comprised of four characters each (Fig. 4a) to a network model with 10 output neurons and 1000 input neurons. Each input neuron generated a 30 ms 10 Hz burst in response to a randomly assigned preferred character (Fig. 4b). This resulted in the formation of three neuron

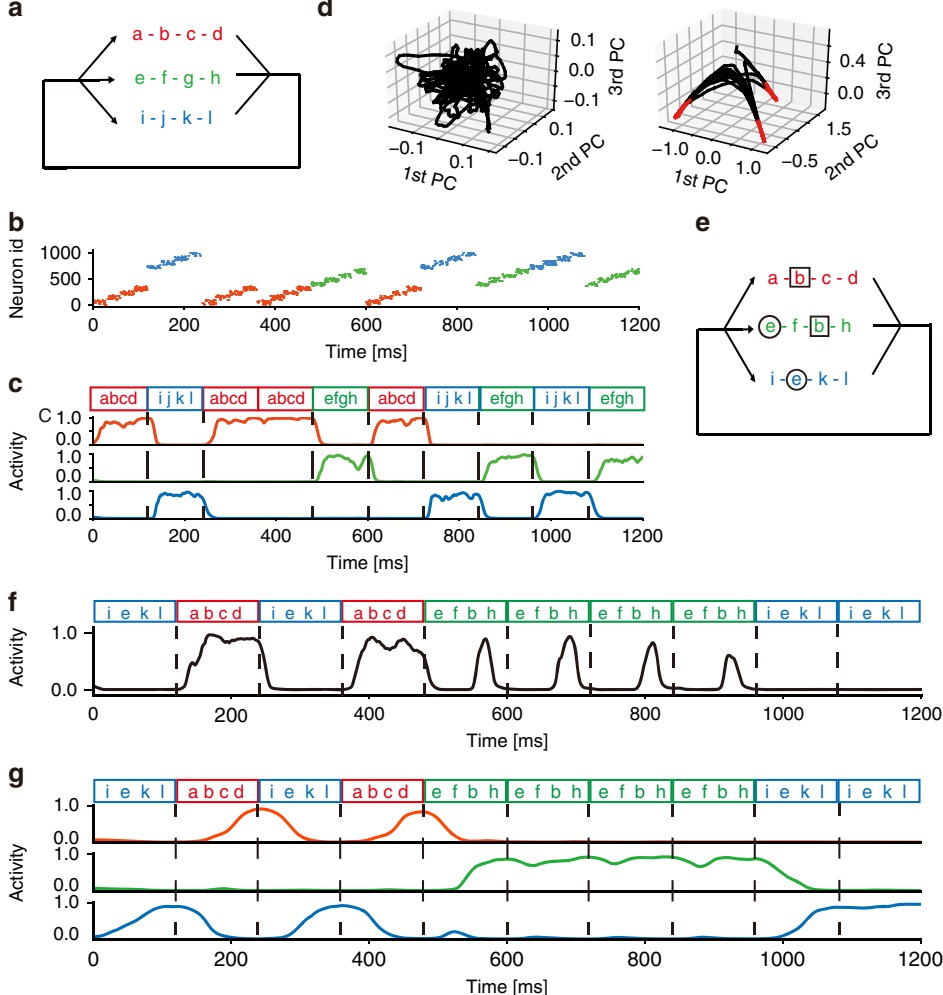

**Fig. 4 Segmentation and concatenation of various sequences.** Ten output neurons were connected with all-to-all inhibitory synapses modifiable by iSTDP.
**a** Three chunks (a-b-c-d [red], e-f-g-h [green], and i-j-k-l [blue]) repeated in the input sequence with equal probabilities. **b** Each input neuron fired at 10 Hz
to encode one of the chunks. Neurons were sorted according to their preferred stimuli. **c** Typical normalized responses of three output neurons are shown
after learning. Colors indicate the epochs of the corresponding chunks. **d** Responses of output neurons were projected onto the three leading principal-
component (PC) vectors before (left) and after (right) learning. More than 99% of the variance was explained by the three PCs. Epochs of highly
normalized responses ($f > 0.8$ in all neurons) are indicated in red. **e** Character b is shared by the red and green chunks, and character e appears in the green
and blue chunks. **f** Response of an output neuron to the overlapping chunks is shown. The time constant of synaptic current was 5 ms. **g** Selective
responses of output neurons to the overlapping chunks are shown when the synaptic time constant was 50 ms.

ensembles which selectively responded to the chunks (Fig. 4c).
Principal-component analysis of the low-dimensional dynamics
of the output neurons revealed the emergence of three chunks
after learning (Fig. 4d). Then, we examined whether the model
can learn partially overlapping chunks. In this case, some
characters were shared between the three chunks (Fig. 4e) and
learning was more difficult than in the previous case. The original
model, with fast synaptic current, failed to generate selective
responses to the chunks (Fig. 4f). However, making the decay
constant of the synaptic current slower (50 ms compared to 5 ms:
see Methods) enabled the model to detect temporal inputs on a
longer timescale and to successfully learn the overlapping chunks
(Fig. 4g). The modified network could also learn chunks even if
they were embedded with distractors, which were random
sequences of arbitrary English characters (a to z) with variable
lengths (3 to 7) (Supplementary Fig. 4). These results suggest
slower synaptic currents such as NMDA receptor-mediated
currents may be important for chunking.

Because the word segmentation shown above is also relatively
easy for other methods[41], we tested our model with more

complex input sequences generated by a random walk on a graph
with a community structure in which the connection of each node
to the other four occurred with an equal probability of 0.25
(Fig. 5a). Here, temporal community is clusters of frequently co-
appearing or mutually predicting stimuli in input sequence. The
detection of this community structure is easy for human subjects
but has proven difficult for conventional machine learning
methods which rely on nonuniform transition probabilities
between elements[5]. Like human subjects, output neurons in our
model easily learned selective responses to members of a temporal
community (Fig. 5b).

The network model could also learn feature detection maps
from continuous sensory streams. All sensory features, either
static or dynamic, arrive at the brain essentially in sequence.
Therefore, we asked whether MRIL enables neural networks to
learn the static features of an input when repeatedly presented in
a temporal sequence. To examine this, we applied a random
sequence of noisy images of oriented bars presented for 40 ms
every 70 ms (Fig. 6a). The output neurons, which initially had
no preferred orientations (Fig. 6b), developed well-defined

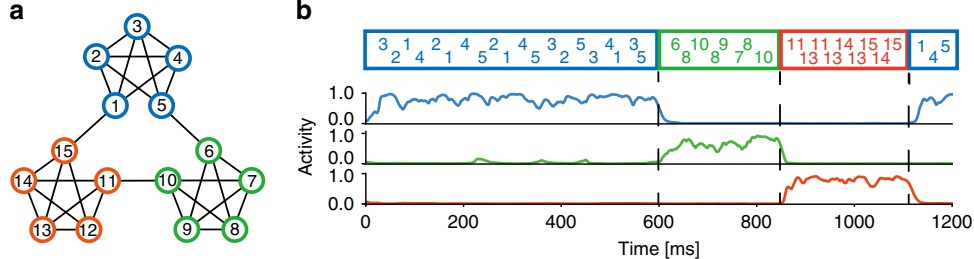

**Fig. 5 Detection of temporal community. a** The input sequence represented a random walk with uniform transition probabilities on a graph with community structure (modified from ref. 5). **b** Normalized responses of three output neurons to input sequences defined in **a** are shown.

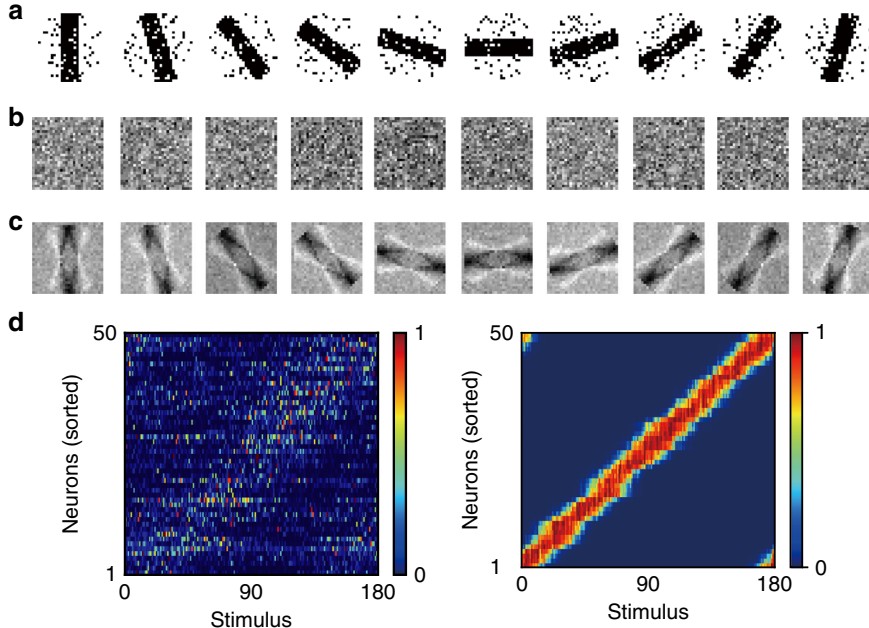

**Fig. 6 Learning an orientation tuning map. a** Examples of noisy images of oriented bars used for training. Each image was presented for 40 ms in a random order with intervals of 30 ms between images. **b**, **c** The feedforward synaptic weights before and after learning are shown for the example stimuli shown in **a**. **d** The responses of all two-compartment neurons before (left) and after (right) learning are shown. The neurons were sorted according to the onset times of responses to their preferred stimuli. See Methods for further details on the simulations.

preferences for specific orientations after learning (Fig. 6c), resembling a visual orientation map (Fig. 6d)[42,43].

**BSS of mutually correlated signals**. The results shown above demonstrate that MRIL successfully chunks a variety of temporal inputs by detecting repeated temporal features. The question then arises whether this ability of the MRIL enables learning of other types of sequence processing tasks. One such task of cognitive and ecological importance is the so-called cocktail party problem[10]. We therefore examined the performance of our network model in the blind separation of mixed signals from multiple sources. BSS is an extensively studied problem in auditory processing[20–22], and various methods have been proposed for mixtures of mutually independent signals. However, methods are limited when the original signals are comprised of mutually correlated signals[23].

We applied MRIL to sound mixtures from two musical instruments, a bassoon and a clarinet (Bach10 Dataset)[44], playing their respective parts of the same score (Fig. 7a) (thus the two sound sources are correlated). A mixed sound followed by the original sounds of the two instruments are presented in Supplementary Audio 1. These mixtures of signals were encoded into irregular spike trains (Fig. 7b), which in turn were applied to

output neurons. After training, these neurons self-organized into two groups, each responding selectively to one of the true sources (Fig. 7c). The original sounds were then decoded from the average firing rates of these subgroups (Supplementary Audios 2 and 3). Although some high-frequency components were lost due to the low-pass filtering effect of the slow membrane dynamics (Supplementary Fig. 5), the decoded sounds are readily comparable to the original sounds. We compared our model with a naive independent-component analysis (FastICA: Supplementary Audio 4)[22,45] and temporal ICA (Second Order Blind Identification or SOBI: Supplementary Audios 5 and 6)[26]. We used the open source software of the SOBI for the comparison (Supplementary Methods). When the source signals are mutually independent, all three methods show excellent performance, although the ICA-based methods slightly outperformed our biology-inspired model (Fig. 7d, top). However, when the source signals are dependent on one another (i.e., mutually correlated), SOBI and our model exhibited significantly better performance than FastICA (Fig. 7d, bottom).

Although SOBI slightly outperformed our model in the present examples, SOBI only poorly performed chunking of the previous sequences of English characters, which our model could easily solve (see Fig. 4). In our simulations, SOBI could not generate

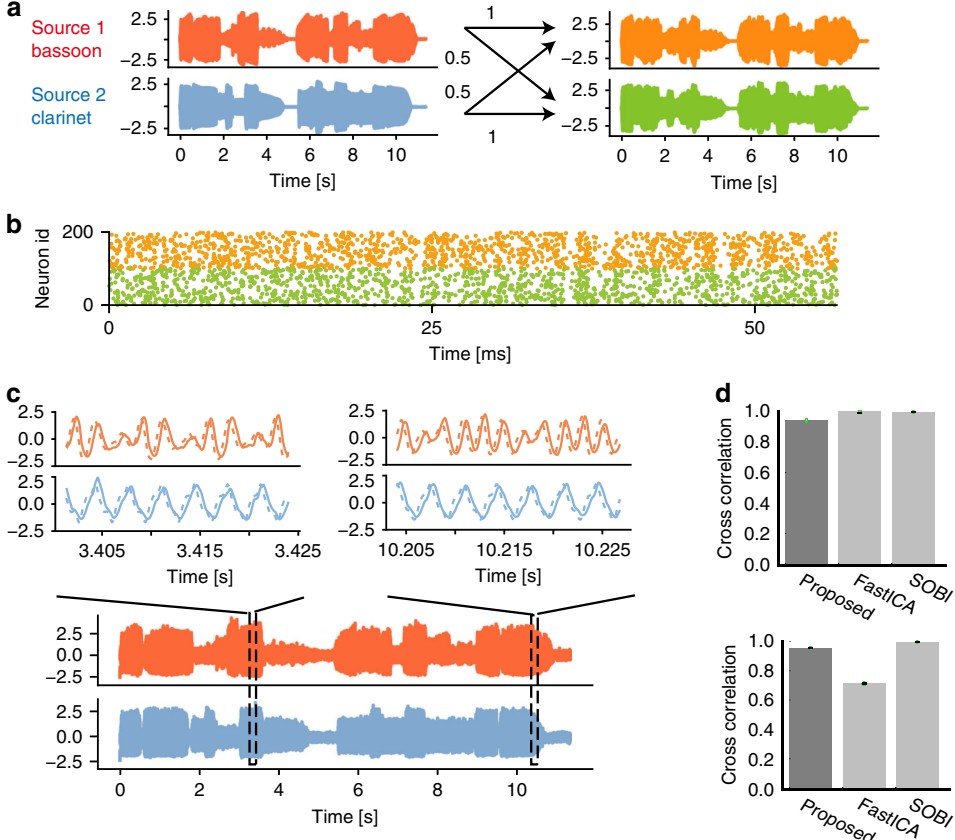

**Fig. 7 BSS of correlated auditory streams. a** Sound waveforms of a bassoon and a clarinet (left) were linearly transformed to two mixed signals (right). The diagonal and off-diagonal elements of the mixing matrix were 1 and 0.5, respectively. **b** Nonstationary Poisson spike trains of 200 input neurons (from a total of 500) are shown. The instantaneous firing rates were proportional to the amplitudes of the mixed signals normalized between 0 Hz and 10 Hz. Each input neuron encodes only one of the two mixed signals. **c** Separated waveforms (bottom) are shown together with magnified versions (top, solid) and true sources (top, dashed). The waveforms were averaged over 20 trials with different realizations of input spike trains and the same initial weights. For the clarity of presentation, error bars are not shown. **d** Cross-correlations between the separated and true sources are compared between our model, FastICA and SOBI for independent (top) and dependent (bottom) auditory signals (see Methods). For each comparison, $n = 20$ independent simulations were performed. Error bars show s.d.s, which were very small. Overlaid green circles are data points.

highly chunk-selective responses (Supplementary Fig. 6a). Rather, most of the units responded to all three chunks in SOBI. We conducted similar analyses for low-pass filtered versions of the input by using different time constants for coarse graining (15, 30 and 50 ms) or the bin width (1 ms or 10 ms), but the essential results remained unchanged (Supplementary Fig. 6b). We also examined SFA, a method known in temporal feature analysis[25], on a similar task by using a Python toolkit[46]. The algorithm failed to generate any stable output when input sequences involved chunks. Detecting a whole chunk and detecting an arbitrary single character cost equally in the objective function of SFA (Methods). Due to this fact, the minimization algorithm of SFA presumably has too many solutions to chunking. Thus, our results demonstrate a virtue of the present brain-inspired model, which exhibits high levels of task performance in a wide range of temporal feature analysis. In addition, the model does not require highly task-specific network architectures.

Finally, we examined the performance of the model by varying the magnitudes of cross-talk noise between the two mixed signals (Methods). We also tested mixed signals which used the same instrument but playing different notes. In all cases, high performance was attained only at an intermediate level of cross-talk noise, implying that performance drops not only for

strong noise but also for weak noise (Fig. 8a, dashed curves). Nevertheless, we could rescue the model from this counter-intuitive defect for weak cross-talk noise by including another noise component (see Methods) in the somatodendritic interaction (Fig. 8a, solid curves). We speculate that the additional noise could suppress learning from harmful interferences between the original signals when both signals were weak. However, this point requires further clarification. We also examined whether the improved model trained on the original signals (i.e., vanishing cross-talk noise) exhibit better performance for other mixtures which were not used in the training. The pre-training actually made the decomposition of unexperienced mixtures easier (Fig. 8b).

## Discussion
Nonlinear Hebbian and generalized STDP algorithms have been used as unsupervised learning rules to perform receptive field development[42,43], ICA[47–49], sparse coding[43], spatio-temporal pattern detection[39,40], or SFA[50]. Our novel algorithm belongs to the same family of methods and is applicable to some classic problems of receptive field development and ICA as well as to the additional problem of 'chunking' as an example task with specific

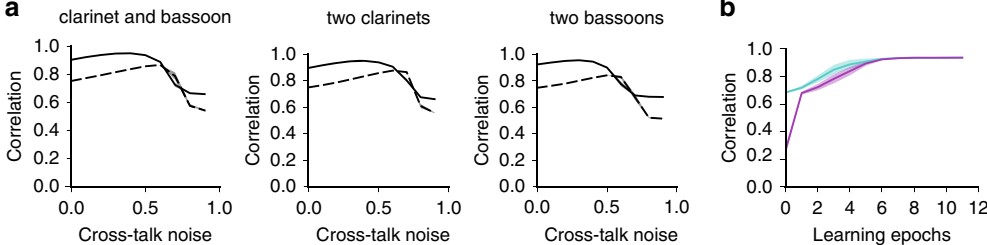

**Fig. 8 Robustness of performance in BSS.** Curves represent the averages over five trials with different initial weights and different realizations of noise, and shaded areas represent s.d. **a** Correlations between the original and separated signals are plotted as a function of cross-talk noise (Methods) for where the mixed signals consist of different (left) or identical (middle and right) instruments. The error bars are very small. **b** Performance is compared between the pre-trained (cyan) and untrained (magenta) network on the original signal. Cross-talk noise in these tests was 0.5. Networks were exposed to 10 s long mixed sounds during each learning epoch and correlations were calculated afterwards.

temporal structure that has traditionally been solved with more specialized algorithms[51,52].

We proposed a learning principle called minimization of regularized information loss (MRIL) which enables the self-supervised learning of recurring temporal features in information streams using a family of competitive networks of two-compartment neuron models. Our model not only performs chunking but also achieves BSS from mixtures of mutually correlated signals. Importantly, although different values of parameters were learned in different tasks, the network structure was essentially the same. It is surprising that simple such neural networks with almost identical circuit structures can perform these broadly different tasks. In particular, our brain-inspired model can solve tasks, e.g., the detection of temporal community structure (Fig. 5) and the BSS of mixed correlated signals (Fig. 7), which conventional models have historically struggled with. To our knowledge, this is the first model to achieve such results on this broad collection of learning tasks.

Our learning rule minimizes the information loss between synaptically-driven dendritic activity and somatic output in the presence of neuronal adaptation. This rule uses mutually inhibiting two-compartment neurons to learn the repetition of temporal activity patterns on a slow timescale (typically, several tens to several hundreds of milliseconds). While the aim of many previous methods for chunking is to predict input sequences[53,54], our model uses a different principle, where system learns to predict its own responses to a given input. MRIL minimizes the discrepancy between input data and output data to produce a predictable low-dimensional representation of high-dimensional input data. This learning continues until an agreement is formed by the somatic output and dendritic input regarding the low-dimensional features (i.e., chunks).

We previously used paired reservoir computing for chunking, where two recurrent networks supervise each other to mimic the partner's responses to a common temporal input[55]. Although that model also learns self-consistency between input data and output data, performance was severely limited since the model required exactly the same number of output neurons as chunks. In contrast, the present model self-organizes output neurons according to the number of temporal features.

Mutual information maximization (MIM) has often been hypothesized to describe the transfer of information between neurons[56], and Hebbian synaptic plasticity may approximately follow MIM[57]. The aim of MRIL differs from MIM; MRIL attempts to detect recurrence, and hence salient, temporal features without considering the other information, whereas MIM ultimately implies that messages are faithfully copied at all layers of hierarchical processing. In other words, MIM does not account

for the compression or abstraction of temporal inputs, whereas MRIL aims to describe how these processes may be executed in the brain and incorporates them into the model. Our results suggest such processes can even occur at the level of single cortical neurons.

Similarly to MRIL, a method called information bottleneck also compresses data streams[58]. The method contains a free parameter to determine the degree of information loss between the original and compressed data. To clarify whether there is a relationship between information bottleneck and the proposed method is an intriguing open question.

A previous model (U-S model)[24] used a learning rule similar to the present one. However, while the somatic response function undergoes activity-history-dependent modulations in our model (see Eqs. 4–7), such modulations were not included in the U-S model. Importantly, our model without these modifications (i.e., the U-S model) could not solve the present unsupervised learning tasks. Networks of the U-S model were shown to perform semi-unsupervised learning, for instance, when recurrent synaptic input was configured as an effective teaching signal to the soma. In contrast, our model indicates the recent history of somatic activity is sufficient for self-supervising the learning of temporal features. We note that the somatic response modifications introduced in this model may be achieved in cortical neurons by local inhibitory circuits[59], the plasticity of intrinsic excitability[60] or neuronal adaptation[16–19].

Dendritic computing has been studied from various viewpoints of neural computing. Memmesheimer et al. derived the capacity of leaky integrate-and-fire neurons to implement desired transformations from streams of input spikes into desired output spike sequences[61]. The capacity was estimated by calculating the available volume of state space for generating the desired spike outputs and an error-correcting supervised learning rule was presented to attain the desired input-output associations (which does not require dendrites). Legenstein and Maass studied the role of nonlinear dendritic processing in performing various logic operations[62]. Their model combines the branch-strength potentiation of dendrites and STDP to discriminate spatial activity patterns represented in presynaptic neuron ensembles. Sacramento et al. used dendrites to implement a classical error back-propagation algorithm for supervised learning where deviations between top-down predictive signals and bottom-up sensory signals provided an error signal[63]. Redundant synaptic connections between neuron pairs have also been utilized to implement a Bayesian filtering algorithm to infer input-output associations in single neurons with realistic dendritic morphology[64]. If such a model includes both the Hebbian learning of synaptic weights and structural plasticity on the dendrites, a small number of

redundant synapses is sufficient for an optimal inference. All of the models of dendritic processing discussed here are biologically more realistic compared to the present model, yet they did not address the ability of neurons with dendrites in analyzing temporal features of information streams.

On the other hand, memory-related sequential activities of hippocampal neurons were modeled in terms of nonlinear amplification of synchronous inputs[65]. Furthermore, the discrimination of sequences on behavioral time scales was recently formulated in terms of the reaction-diffusion processes triggered by sequential inputs along dendrites[66]. While these processes were implemented in morphologically realistic neuron models, whether such models can perform complex temporal feature analyses is yet to be clarified. Hawkins and Ahmad modeled sequence processing in a cortical microcircuit model of formal neurons, each of which receives top-down feedback inputs on apical synapses, feedforward inputs on proximal synapses and lateral inputs from nearby neurons on multiple dendritic segments[67]. Through coincidence detection and segment-basis Hebbian learning, the network learns to recognize sparse activity patterns and to predict next spikes in an input sequence. While their model emphasizes the role of dendrites and cortical microcircuit structure in predicting spike sequences, our model demonstrates the ability of single neurons with dendrites to learn recurring temporal input patterns.

Determining which neuron or synapse should be credited for learning a desired output in a hierarchical neural circuit is a difficult problem. Solutions to this 'credit assignment problem' require feedback signals to neurons or synapses. In cortical pyramidal neurons, feedforward sensory data is thought to be received at the basal dendrites while feedback credit information is received at apical dendrites. It was recently argued that the spatial separation between the two pathways enables these neurons to solve the credit assignment problem through dendritic computing[68]. The current version of our model does not solve the credit assignment problem, and this problem arises on multiple timescales in hierarchical brain computation. How morphologically complex neurons implement the proposed temporal feature analysis and how this analysis helps the brain to solve hierarchically organized credit assignment problems are intriguing open questions.

## Methods

**Neural network model.** Each output neuron has two compartments—somatic and dendritic. The dendritic membrane potential of output neuron $i \in \{1, 2, \ldots, N_{\text{out}}\}$ is calculated as

$$v_i(t) = \sum_j w_{ij} e_j(t), \quad (1)$$

where $w_{ij}$ is the synaptic weight between output neuron $i$ and input neuron $j$. The variable $e_j$ stands for the unit postsynaptic potential induced by neuron $j$ and is described later. The somatic activity integrates the dendritic potential, and it evolves as

$$\dot{u}_i(t) = -\frac{1}{\tau} u_i(t) + g_{\text{D}}[-u_i(t) + v_i(t)] - \sum_j G_{ij} \phi^{\text{som}}(u_j(t))/\phi_0, \quad (2)$$

where $\tau = 15$ ms and the conductance between the two compartments is $g_{\text{D}} = 0.7$. The last term describes lateral inhibition with synaptic weights $G_{ij}$ ($\geq 0$). We calculated the inhibitory input in terms of the firing rates of output neurons. However, as explained below, spike trains of these neurons were also generated for simulating modifications of $G_{ij}$ by spike-timing-dependent plasticity. We assume that the soma of neuron $i$ generates a Poisson spike train with the instantaneous firing rate $\phi_i^{\text{som}}(u_i(t))$ in terms of the nonlinear response function

$$\phi_i^{\text{som}}(u_i) = \phi_0 [1 + \exp(\beta_i(-u_i + \theta_i))]^{-1}. \quad (3)$$

The parameters $\beta_i$ and $\theta_i$ are defined as follows:

$$\beta_i = \sigma_i(t)^{-1} \beta_0, \quad (4)$$

$$\theta_i = \mu_i(t) + \sigma_i(t) \theta_0, \quad (5)$$

where $\mu_i(t)$ and $\sigma_i(t)$ are the mean and variance of the membrane potential, respectively, over a sufficiently long period $t_0$:

$$\mu_i(t) = \frac{1}{t_0} \int_{t-t_0}^t u_i(t') dt', \quad (6)$$

$$\sigma_i(t) = \sqrt{\frac{1}{t_0} \int_{t-t_0}^t u_i(t')^2 dt' - \mu_i(t)^2}. \quad (7)$$

We set $\beta_0 = 5$ throughout this study, but the values of $\phi_0$ and $\theta_0$ are task-depend (Supplementary Methods).

We note that the slope of nonlinearity $\beta_i$ and the threshold value $\theta_i$ are modified as the values of $\mu_i$ and $\sigma_i$ change during learning. As described below, the online modifications of the somatic response function maintain the dynamic range of output firing rate within a certain range. To see this, we use Eqs. (4) and (5) to obtain

$$\phi_i^{\text{som}}(u_i)/\phi_0 = [1 + \exp(\beta_0(-\hat{u}_i + \theta_0))]^{-1} = \hat{\phi}(\hat{u}_i)/\phi_0,$$

where $\hat{\phi}(x) = \phi_0[1 + \exp(\beta_0(-x + \theta_0))]^{-1}$ and $\hat{u}_i(t) \equiv (u_i(t) - \mu_i(t))/\sigma_i(t)$. As the mean of $\hat{u}_i(t)$ is constrained to be zero, the above equation implies that $\phi_i^{\text{som}}(u_i)/\phi_0$ is also constrained around $[1 + e^{\beta_0 \theta_0}]^{-1}$ with fluctuations of $O(1)$. Thus, the somatic activity does not saturate.

In our model, sensory information given to the network is first encoded into Poisson spike trains. Input neuron $i \in \{1, 2, \ldots, N_{\text{in}}\}$ generates a Poisson spike train

$$X_i(t) = \sum_q \delta(t - t_{i,q}), \quad (8)$$

where $\delta$ is the Dirac' delta function and $t_{i,q}$ denotes the time of the $q$-th spike of input neuron $i$. The presynaptic spikes induce the following synaptic current $I_i(t)$:

$$\tau_{\text{syn}} \dot{I}_i = -I_i + \frac{1}{\tau} X_i, \quad (9)$$

where the synaptic time constant $\tau_{\text{syn}} = 5$ ms ($\tau_{\text{syn}} = 50$ ms in Fig. 4g and Supplementary Fig. 4c). The synaptic currents in turn evoke a postsynaptic potential $e_i(t)$ as

$$\dot{e}_i = -\frac{e_i}{\tau} + e_0 I_i. \quad (10)$$

The unit amplitude of postsynaptic potentials is given as $e_0 = 25$ in all simulations.

**Optimal learning rule for MRIL.** To extract the characteristic features of the temporal input, our model compresses the high dimensional data carried by the input sequence onto a low dimensional manifold of neural dynamics. The model performs this by modifying the weights of dendritic synapses to minimize the time-averaged mismatch between the somatic and dendritic activities over a certain interval $[0, T]$. In a stationary state, the somatic membrane potential $u_i(t)$ of a two-compartment model can be described as an attenuated version $v_i^*(t)$ of the dendritic membrane potential with an attenuation factor $\alpha = g_{\text{D}}/(g_{\text{D}} + g_{\text{L}})$, where $g_{\text{L}} = \tau^{-1}$[24]. Though we deal with time-dependent stimuli in our model, we compare the attenuated dendritic membrane potential with the somatic membrane potential at each time point. This comparison, however, is not drawn directly on the level of the membrane potentials but on the level of the two Poissonian spike distributions with rates $\phi_i^{\text{som}}(u(t))$ and $\hat{\phi}(v_i^*(t))$, respectively, which would be generated if both soma and dendrite were able to emit spikes independently. The function $\hat{\phi}(v_i^*(t))$ can also be regarded as a nonlinear-filtered version of the attenuated dendritic membrane potential[69].

Explicitly representing the dependency of $u_i$ and $v_i^*$ on $\mathbf{X}$, we define the cost function for synaptic weights $\mathbf{w}$ as

$$E(\mathbf{w}) = \int_{\Omega_{\mathbf{X}}} d\mathbf{X} P^*(\mathbf{X}) \int_0^T dt \sum_i \text{D}_{\text{KL}}\left[\phi_i^{\text{som}}(u_i(t; \mathbf{X})) \phi^{\text{dend}}(v_i^*(t; \mathbf{X}))\right], \quad (11)$$

where $P^*(\mathbf{X})$ stands for the true distribution of input spike trains, $\Omega_{\mathbf{X}}$ for the space spanned by all possible combinations of input spike trains, and $\text{D}_{\text{KL}}$ for the KL-divergence between the two Poisson distributions:

$$\begin{aligned} &\text{D}_{\text{KL}}\left[\phi_i^{\text{som}}(u_i(t; \mathbf{X})) \phi^{\text{dend}}(v_i^*(t; \mathbf{X}))\right] \\ &\equiv \phi_i^{\text{som}}(u_i(t; \mathbf{X})) \log \frac{\phi_i^{\text{som}}(u_i(t; \mathbf{X}))}{\phi^{\text{dend}}(v_i^*(t; \mathbf{X}))} + \phi^{\text{dend}}(v_i^*(t; \mathbf{X})) \\ &\quad - \phi_i^{\text{som}}(u_i(t; \mathbf{X})) \end{aligned}$$

with $\phi^{\text{dend}}(x) = \phi_0[1 + \exp(\beta_0(-x + \theta_0))]^{-1}$. Note that unlike the somatic response function $\phi_i^{\text{som}}$, of which the values of $\beta_i$ and $\theta_i$ are neuron-dependent, the function $\phi^{\text{dend}}$ is common to all neurons.

We minimize the cost function (i.e., the averaged KL-divergence) with respect to $\mathbf{w}$ such that the responses of the two compartments become consistent with each other. Thus, the unsupervised learning rule of somatodendritic consistency check resolves the discrepancy between the somatic and dendritic responses to temporal

input. Similar to reference[24], we search for the optimal weight matrix by gradient descent as

$$
\begin{aligned}
\Delta w_{ij} &\propto -\frac{\partial}{\partial w_{ij}} E \\
&= -\frac{\partial}{\partial w_{ij}} \int_{\Omega_{\mathbf{X}}} d\mathbf{X} P^*(\mathbf{X}) \int_0^T dt \sum_{i'} \mathrm{D_{KL}} \left[ \phi_{i'}^{\mathrm{som}}(u_{i'}(t; \mathbf{X})) \phi^{\mathrm{dend}}(v_{i'}^*(t; \mathbf{X})) \right] \\
&= -\int_{\Omega_{\mathbf{X}}} d\mathbf{X} P^*(\mathbf{X}) \int_0^T dt \frac{\partial}{\partial w_{ij}} \left[ \phi_i^{\mathrm{som}}(u_i(t; \mathbf{X})) \log \frac{\phi_i^{\mathrm{som}}(u_i(t; \mathbf{X}))}{\phi^{\mathrm{dend}}(v_i^*(t; \mathbf{X}))} + \phi^{\mathrm{dend}}(v_i^*(t; \mathbf{X})) - \phi_i^{\mathrm{som}}(u_i(t; \mathbf{X})) \right] \\
&= \int_{\Omega_{\mathbf{X}}} d\mathbf{X} P^*(\mathbf{X}) \int_0^T dt \frac{d\log(\phi^{\mathrm{dend}}(v_i^*(t; \mathbf{X})))}{\partial w_{ij}} \left[ \phi_i^{\mathrm{som}}(u_i(t; \mathbf{X})) - \phi^{\mathrm{dend}}(v_i^*(t; \mathbf{X})) \right]
\end{aligned}
\tag{13}
$$

Note that the identity $d\phi^{\mathrm{dend}}(x)/dx = \phi^{\mathrm{dend}}(x) d\log \phi^{\mathrm{dend}}(x)/dx$ was used in deriving the last expression. Since $v_i^*(t) = \alpha \sum_j w_{ij} e_j(t)$, the local learning rule is written in a vector form as

$$
\Delta \mathbf{w}_i \propto \int_{\Omega_{\mathbf{X}}} d\mathbf{X} P^*(\mathbf{X}) \int_0^T dt \psi(v_i^*(t; \mathbf{X})) \left[ \phi_i^{\mathrm{som}}(u_i(t; \mathbf{X})) - \phi^{\mathrm{dend}}(v_i^*(t; \mathbf{X})) \right] e(t; \mathbf{X}),
\tag{14}
$$

where $\mathbf{w}_i = [w_{i1}, \cdots w_{iN_{\mathrm{in}}}]$ and the function $\psi(x)$ is defined as

$$
\psi(x) = \frac{d}{dx} \log(\phi^{\mathrm{dend}}(x)).
\tag{15}
$$

Note that the $i$-dependence of $\Delta \mathbf{w}_i$ arises in our network model from activity-dependent modifications of recurrent inhibitory connections among output neurons (see Eq. 2). The inhibitory connections are modifiable by STDP (see Fig. 2b).

In all simulations, we added the regularization term $-\gamma \mathbf{w}_i$ to Eq. (14) to prevent the diverging growth of synaptic weights. Thus, the following online learning rule was used:

$$
\mathbf{w}_i(t) = \eta \left\{ \psi(v_i^*(t)) \left[ \left\{ \phi_i^{\mathrm{som}}(u_i(t)) - \phi^{\mathrm{dend}}(v_i^*(t)) \right\} / \phi_0 \right] \mathbf{e}(t) - \gamma \mathbf{w}_i \right\},
\tag{16}
$$

where $\eta$ is the learning rate. The parameter $\gamma$ controls the strength of regularization and was adjusted in a task-dependent manner. The initial values of $\mathbf{w}$ were generated by a Gaussian distribution with mean zero and standard deviation $1/\sqrt{N_{\mathrm{in}}}$. Note that the above learning rule coincides with the Bienenstock-Cooper-Munro (BCM) theory except for a sign difference[70]. In BCM theory, the threshold between potentiation and depression is an unstable fixed point while in our model this point is a stable fixed point. However, as shown previously, the online modifications given in Eqs. (4)–(7) prevent the function $\phi_i^{\mathrm{som}}(u_i(t))$ from coinciding with $\phi^{\mathrm{dend}}(v_i^*(t))$. This in turn prevents a trivial fixed point $\mathbf{w} = 0$ of Eq. (16). We note that the online modifications of somatic response function play a similar role to the standardization method to avoid a trivial solution in the SFA of temporal input[25].

A similar learning rule to Eq. (16) was previously considered in a supervised learning model in which the average surprise of somatic spike output driven by dendritic synaptic input and a teaching signal given to the soma was minimized[24]. In this analogy, our learning rule may be interpreted as self-consistent surprise minimization in which the teaching signal itself is provided by the somatic response to make the learning rule for two-compartment neurons unsupervised. This summarizes the essential difference between our model and the previous model.

**Improved learning rule with additional noise.** In Fig. 8, we included an additional noise term at each time step of learning as follows:

$$
\mathbf{w}_i(t) = \eta \left\{ \psi(v_i^*(t)) \left[ \left\{ f(\phi_i^{\mathrm{som}} + \phi_0 g \xi_i) - \phi^{\mathrm{dend}}(v_i^*(t)) \right\} / \phi_0 \right] \mathbf{e}(t) - \gamma \mathbf{w}_i \right\},
\tag{17}
$$

where $\xi_i$ is a random variable obeying a normal distribution. The parameter $g$ controls the strength of the noise, and we set $g = 0.6$ in Fig. 8. The piecewise linear function $f$ is defined as

$$
f(x) = \begin{cases} 0 & x < 0 \\ x & 0 \le x < \phi_0. \\ \phi_0 & x \ge \phi_0 \end{cases}
\tag{18}
$$

Negative signals should be eliminated to suppress the learning during noise-dominant epochs.

**Inhibitory plasticity.** We modified lateral inhibitory connections through a symmetric anti-Hebbian STDP: if a pair of presynaptic and postsynaptic spikes occur at the times $t_{\mathrm{pre}}$ and $t_{\mathrm{post}}$, respectively, the weight changes were calculated as

$$
\Delta G_{ij} = C_{\mathrm{p}} \exp\left(-\frac{|t_{\mathrm{pre}} - t_{\mathrm{post}}|}{\tau_{\mathrm{p}}}\right) - C_{\mathrm{d}} \exp\left(-\frac{|t_{\mathrm{pre}} - t_{\mathrm{post}}|}{\tau_{\mathrm{d}}}\right),
\tag{19}
$$

where $\tau_{\mathrm{p}}$ and $\tau_{\mathrm{d}}$ are the decay constants of LTP and LTD, respectively[35,36].

Typically, $\tau_{\mathrm{p}} = 40$ ms, $\tau_{\mathrm{d}} = 20$ ms, $C_{\mathrm{p}} = 0.00525$ and $C_{\mathrm{d}} = 0.0105$. Inhibitory weights $G_{ij}$ were modified between zero and an upper bound $G_{\mathrm{max}}(\propto 1/\sqrt{N_{\mathrm{out}}})$.

**Evaluation of the degree of independency between signals.** ICA was not valid for the auditory signals used for the simulations of BSS. This was because the signals were not independent. In addition to the standard correlations between two analog signals, negentropy ($\ge 0$) was used to evaluate the independency of signals. Negentropy measures the deviation of a target distribution from a Gaussian distribution: negentropy vanishes if the target distribution is Gaussian but otherwise takes a positive value; the larger the deviation is, the larger the value of negentropy is. The calculation of negentropy $J(Y)$ for the statistical variable $Y$ requires the true distribution, but it is unknown in the present study. Therefore, we made the following approximation in the evaluation of $J(Y)$ using a function $Q$:

$$
J(Y) \propto [E(Q(Y)) - E(Q(\rho))]^2,
\tag{20}
$$

where $E(x)$ refers to the expectation value of $x$ and $\rho$ obeys a Gaussian distribution. Typically, the logarithm of hyperbolic cosine function is used for $Q$[49]:

$$
Q(u) = \frac{1}{a} \log \cosh(au),
\tag{21}
$$

where $1 \le a \le 2$. In this study, we set as $a = 1$.

**Cross-talk noise.** In Fig. 8, we mixed the original signals $X_1(t)$ and $X_2(t)$ as follows:

$$
\begin{pmatrix} \cos\theta & \sin\theta \\ \sin\theta & \cos\theta \end{pmatrix} \begin{pmatrix} X_1(t) \\ X_2(t) \end{pmatrix}.
\tag{22}
$$

Then, the cross-talk noise between the two mixed signals was defined as $\tan\theta$. The mixed signals coincide with the original signals at $\theta = 0$, while the two mixtures are identical at $\theta = \frac{\pi}{4}$ and BSS is a single-source separation problem.

**Chunking of character sequences by SOBI and SFA.** In Supplementary Fig. 6a, we applied SOBI to the same sequential input as used in Fig. 5a. In the simulations of SOBI, the number of input units was set equal to the number of characters in the sequence, and each unit takes the value 1 when the corresponding character appears in input and 0 otherwise. In (B), we low-pass filtered a raw input with the time constant of 50 ms, and then resampled the filtered input with the time step of 10 ms priori to the application of SOBI. We also employed different values of the time constant (15 and 30 ms) and time step (1 ms), but these modifications did not change the essential results.

Denoting the observed time-series data at time $t$ and the output of the $j$-th unit as $\mathbf{X}_t$ and $y_{j,t} = g_j(\mathbf{X}_t)$, respectively, we can describe the outline of SFA as follows. The objective of SFA is to minimize the following quantity ($\Delta$-value):

$$
\Delta(y_{j,t}) \equiv \left\langle \dot{y}_{j,t}^2 \right\rangle_t,
\tag{23}
$$

where $\langle \cdot \rangle_t$ denotes the averaging over time, under the following three constraints:

$$
\left\langle y_{j,t} \right\rangle_t = 0,
\tag{24}
$$

$$
\left\langle y_{j,t}^2 \right\rangle_t = 1,
\tag{25}
$$

$$
\left\langle y_{i,t} y_{j,t} \right\rangle_t = 0.
\tag{26}
$$

In other words, we should find out the scalar function $g_j(\mathbf{X}_t)$ that minimizes the time derivative of the latent variable $y_{j,t}$. Then, the latent variable $y_{j,t}$ that minimizes the $\Delta$-value is called the slow feature of $\mathbf{X}_t$. Equations (24) and (25) prevent a trivial solution as in our model, and Eq. (26) deccorelates the outputs of different units. We applied SFA[25] to the same input sequence as used in Fig. 5a. However, the results are not shown as the algorithm failed to generate outputs within a reasonably long simulation time.

**Details of simulations.** Additional technical details of simulations and the values of model parameters used in the figures are given in Supplementary Methods.

**Reporting summary.** Further information on research design is available in the Nature Research Reporting Summary linked to this article.

## Data availability
All numerical datasets necessary to replicate the results shown in this article can easily be generated by numerical simulations with the software code provided below. No datasets were generated during this study.

## Code availability
All codes were written in Python3 with numpy 1.17.3 and scipy 0.18.1. Example program codes used for the present numerical simulations and data analysis are available at https://github.com/ToshitakeAsabuki/MRIL_codes.

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

## Acknowledgements

The authors express their sincere thanks to Shun-ichi Amari for stimulating discussion about the proposed learning rule and to Shigeyoshi Fujisawa, Joshua Johansen and Yukiko Goda for their valuable comments on our manuscript. The authors also thank Yuanchieh Ling and Thomas Burns for technical assistance. This work was partly supported by KAKENHI (nos. 17H06036, 18H05213 and 19H04994) to T.F. T.A. was supported by the Junior Research Associate program of RIKEN and the SRS Research Assistantship of OIST.

## Author contributions

T.F. and T.A. designed the study and wrote the manuscript, and T.A. performed numerical simulations.

## Competing interests

The authors declare no competing interests.
