## [Peer Review File · Nature Communications]

Reviewers' Comments:

Reviewer #1:

Remarks to the Author:

In this paper the authors claim that a 2 compartment neuron, that minimizes the average surprise between its inputs and its own output, enables learning sequences in an unsupervised manner. They claim that a network of such neurons, connected via mutual inhibition, can perform chunking and BSS. There are a number of concerns about the claims made in the paper and the exact functionality of the network, as outlined below. In addition the authors should discuss and compare their technique against existing techniques (see below). Finally, there are many small grammatical inconsistencies and stylistic issues. They could benefit from an editing pass by a native English speaker used to scientific prose. These concerns should be addressed before the report is published.

The abstract, which should be a standalone piece of text, was unclear in a few places. 1) in the abstract they state, "We show that the same networks of the dendritic...". It is not clear which network the statement is referring to - no networks are mentioned in the prior few sentences. The rest of the abstract is about the power of single neurons, so the relevance of networks is unclear in the abstract. 2) The abstract states "...including the chunking of community structure in temporal sequences" - it's not clear what community they are referring to. It might be clearer to just say "including the chunking of structure in temporal sequences".

Fig 2 - how long did it take the network to create three distinct assemblies? The learning curve is not discussed.

Page 8 - "that human subjects can detect the recurrence of frozen noise patterns embedded in a noisy auditory" - but those studies did not require such extensive learning. In Agus et al., subjects learned the noise after 4 presentations of noise patterns. The learning mechanisms therefore might be significantly different from the method presented here. The authors should comment on this discrepancy in the paper.

Fig 3 - how is this sequence different from showing three fixed random chunks? What is the significance of the individual letters? The graph structure shows 5 nodes for each chunk, but there are four letters. There are no links between 11 and 15, and 10 and 6 - why is that?

Fig 3 - what is the significance of the PCA analysis? The significance is not described in the text - there is only one sentence on it.

Fig 3 - the characters don't overlap. Without overlap the learning is fairly trivial, and almost any system can learn the sequence. What would happen if some of the characters were shared between the chunks, such as "abcd", "efbh", and "iekl"?

Fig 3 - Each "input sequence" is a sequence of random patterns from the input. From Fig 3 (E and F), it looks like the order of the input neuron activity doesn't really matter. Each neuron tends to be active anytime within its preferred "community structure". So why do the authors state the neurons are learning sequence structures? Aren't they just learning to respond to the set of random input patterns that are unique to each chunk? If it truly was learning a temporal sequence, then disrupting the temporal order should cause failures in recognition. This reviewer is unclear as to what the authors mean when they state that their algorithm enables the "self-supervised learning of multiple recurring temporal features" (discussion, page 13).

Fig 4 - As I understand it, there were only two audio files, one for each instrument. The network was trained on a single mixture for 60 repetitions and then tested on that same mixture. This is an extremely restricted experiment, and does not justify the claims made by the authors that their model "achieves BSS from mixtures of correlated signals". Examples of more stringent tests are below:

1) Clarinet and bassoon seem to have different average frequencies, and as such it is hard to make any claims that the network is picking up on correlations. Is the network simply picking up on higher vs lower frequencies? What happens if the audio samples have similar frequencies? For example, a clarinet playing two different tunes?

2) What happens if you test on a mixture that the network was not trained on exhaustively? This is a more realistic scenario.

3) The authors linearly mixed two sources using a single fixed mixing matrix. This seems unrealistic since one source is weighted twice as much as the other. What happens if other coefficients are used, such as 0.5 and 0.5?

Relationship to other work:

1) BSS: On page 11 - The authors state that, "to our knowledge, there are no effective methods for separating mixtures of dependent or correlated signals." This is a strong claim. In fact, there is a long literature on machine learning methods for BSS. Two examples, focused on single channel source separation, are:

He, P., She, T., Li, W., & Yuan, W. (2018). Single channel blind source separation on the instantaneous mixed signal of multiple dynamic sources. *Mechanical Systems and Signal Processing*, 113, 22–35. <http://doi.org/10.1016/J.YMSSP.2017.04.004>

Hershey, J. R., Chen, Z., Roux, J. Le, & Watanabe, S. (2015). Deep clustering: Discriminative embeddings for segmentation and separation. Retrieved from <http://arxiv.org/abs/1508.04306>

They should perform a more detailed comparison to existing methods, rather than just a naive application of ICA. Python source code for the second reference above is available here: <https://github.com/chaodengusc/DeWave>

2) Dendritic learning: They should compare their learning techniques and discuss the following papers which model either dendritic learning and/or sequences in spiking networks:

Legenstein, R., & Maass, W. (2011). Branch-specific plasticity enables self-organization of nonlinear computation in single neurons. *The Journal of Neuroscience: The Official Journal of the Society for Neuroscience*, 31(30), 10787–802. <http://doi.org/10.1523/JNEUROSCI.5684-10.2011>

Memmesheimer, R. M., Rubin, R., Ölveczky, B., and Sompolinsky, H. (2014). Learning precisely timed spikes. *Neuron* 82, 925–938. doi: 10.1016/j.neuron.2014.03.026

Hawkins, J., & Ahmad, S. (2016). Why Neurons Have Thousands of Synapses, a Theory of Sequence Memory in Neocortex. *Frontiers in Neural Circuits*, 10(23), 1–13. <http://doi.org/10.3389/fncir.2016.00023>

Jahnke, S., Timme, M., and Memmesheimer, R.-M. (2015). A unified dynamic model for learning,

replay, and sharp-wave/ripples. *J. Neurosci.* 35, 16236–16258. doi: 10.1523/JNEUROSCI.3977-14.2015

Richards, B. A., & Lillicrap, T. P. (2019). Dendritic solutions to the credit assignment problem. *Current Opinion in Neurobiology*, 54, 28–36. <http://doi.org/10.1016/J.CONB.2018.08.003>

Sacramento, J., Costa, R. P., Bengio, Y., & Senn, W. (2017). Dendritic error backpropagation in deep cortical microcircuits. Retrieved from <http://arxiv.org/abs/1801.00062>

"We model this learning process as a self-supervision by single neurons which minimizes time-averaged surprise about spike output for given dendritic activity." "activity in the dendritic compartment, driven by external inputs, predicts somatic spike responses" - how does this compare to Hawkins & Ahmad where dendritic events are predictions of a neuron's output? They too use bAP's to train the dendritic compartment and leads to an unsupervised learning system.

Reviewer #2:

Remarks to the Author:

It is well known that nonlinear Hebbian and generalized STDP algorithms can be used as unsupervised learning rules to perform ICA, sparse coding, spatio-temporal pattern detection, or slow feature analysis (SFA). The paper of Asabuki and Fukai adds another algorithm in this space. Specifically, the novel algorithm combines insights from SFA (ref 42) with insights from the learning rule of Urbanczik and Senn (ref 20).

The paper has a nice story line starting with two-compartment neurons and surprise minimization leading to a learning rule similar to the one of Urbanczik-Senn. However, this story line is misleading, because the mathematical derivations that the authors present to derive the learning rule (Eq. 19) do not support this claim. The main steps for the derivations follow the arguments of Urbanczik and Senn, starting from a slightly different objective function. Yet, the objective function DOES NOT give the desired learning rule, because the essential insight enters heuristically (and through the backdoor): the \hat{u} variable is set to unit variance. This aspect is inherited from SFA (as it is correctly acknowledged), but in contrast to SFA where this aspect is part of the objective function and its constraints, it comes in the present paper completely out of the blue.

My main concerns are made more precise in the following comments.

MAJOR COMMENTS

1) The algorithm has its merits and may very well be that the new learning rule is stronger than existing unsupervised rules. However, in order to show this, a comparison with other learning rules must be provided.

For ICA (Fig 4), the comparison should be with other TEMPORAL ICA rules (such as Molgedy and Schuster, see refs 1-6 below), and not the ICA rules based on spatial higher-order correlations cited in the paper. These temporal ICA rules have a known link to asymmetric Hebbian learning (such as STDP).

Also, a comparison with SFA should be done on all three paradigms (Figs 1-4), for example using the SFA version of Sprekeler et al. 2014. (refs 6-8). These SFA rules can be implemented by local Hebbian

learning rules and are thus at the same level as the rule in the present manuscript.

For spatio-temporal patterns (Figs 1 and 2) the comparison should be with other unsupervised rules that can be used on the same task, such as Masquelier and Thorpe or Nessler et al. (refs 9, 10).

2) The theoretical derivation must be completely rewritten. The current formulation via surprise is completely misleading, because if you were to optimize average surprise WITHOUT the condition on \hat{u} , you would find the constant solution as the best one.

The aim is to arrive at Eq. (19). Question: What is the objective function such that optimization yields Eq. (19)? Somehow the constraint on \hat{u} must become part of the derivation - otherwise the derivation remains a copy of the theory of Urbanczik and Senn. My guess is that similar to variational autoencoders (I mean the encoding step, not the decoding step) you could require that the distribution of \hat{u} has unit variance and zero mean, for example. Can you define a KL divergence so that v will approach \hat{u} ? Then optimize this and arrive directly at Eq. (19)? It might not be easy, but somehow the constraints on \hat{u} must be part of the formalism.

3) It is not even clear that you need a two-compartment neuron.

You could also consider \hat{u} as some low-pass filtered version of v^* that is represented in the Ca-dynamics or similar. Define three online low-pass variables for v^* , $(v^*)^2$, and a normalized combination for these (your \hat{u}).

4) For an additional point of view of your novel learning rule, you may consider the following analogy.

In a general BCM-type learning rule, the weight updates are proportional to

$$\frac{d}{dt}w_{ij} = \eta [\Psi(v^*_i)[\phi(v^*_i) - \theta(v^*_{bar}_i)] e_j - \gamma w_{ij}]$$

where $v^*_{bar}_i$ is some running average of the postsynaptic activity v^*_i . See Bienenstock et al. 1982.

I can rewrite your Eq. (19) similarly as

$$\frac{d}{dt}w_{ij} = \eta [\theta(v^*_{bar}_i) - \phi(v^*_i)] e_j - \gamma w_{ij}]$$

where $v^*_{bar}_i$ is the normalized running average of the postsynaptic activity v^*_i (your \hat{u}).

What are the main differences?

(i) There is a sign difference. In BCM $\theta(v^*_{bar})$ is an UNSTABLE fixed point. This instability is necessary to create sensitivity to fluctuations that the learning rule needs to enhance to form receptive fields or find independent components. If you switch the sign, this turns into a stable fixed point. Normally this would be a boring fixed point because no receptive fields or independent components can be learned, the neuron just normalises its rate to a desired mean. However:

(ii) the second difference is that v^*_{bar} in YOUR case is normalised so as to have a standard deviation of one (like in SFA). So the learning rule cannot converge to a fixed point, but must adapt the weights so that v^* tries to follow the normalized fluctuations of v^*_{bar} . Thus the sensitivity to fluctuations comes no longer from the learning rule we see in your Eq. (19), but from the definition of v^*_{bar} as a NORMALIZED running average.

Maybe these analogies help to rewrite the paper.

5) The story-line of the paper has to be rewritten from scratch. In my opinion the learning rule has nothing to do with surprise, but a lot with making the v -distribution close to the u -hat distribution which is BY DEFINITION normalized. See the points 2-4 above.

6) Unsupervised versus supervised; generative model versus recognition model.

You seem to imply that Urbanczik and Senn only use a supervised paradigm. However, they already studied an unsupervised paradigm with lateral interactions (but not the single neuron). So, explain better please what is different in your case.

Moreover you seem to imply that sparse coding or ICA or variational autoencoders are different because they aim at a reconstruction of the original signal. However, the Dayan-Abbott textbook (see Chapter 10, and table in appendix 10.6) explains that the generative model view ('reconstruction') and the recognition view ('decoding') and the Hebbian learning rule view (online local algorithm) are just three different, but equivalent, views of the SAME learning algorithm. This parallelism has been rediscovered and discussed by many people over the last 20 years. For example, one can apply the Olshausen learning rule in order to learn a sparse representation WITHOUT the framework of stimulus reconstruction. And follow-up work from Friedrich Sommer in the Olshausen group has shown that one can also just see it as an unsupervised learning rule, not even bothering about the coding.

MINOR COMMENTS:

p.5, lines 20-25. As you know, real neurons are either excitatory or inhibitory. Unclear whether you think of this learning rule like a rule for the inhibitory synapses (see review of Sprekeler et al. 2013) or of the excitatory neurons onto inhibitory neurons (as your reference seems to suggest).

p 15. Eq. 2. I understand from the main text that you use spiking neurons. But then the last term in Eq. 2 should be a sum of the spike trains of other neurons, and not over their firing rates.

p15. Eq. 3. I understand that spikes are in the end drawn from $\phi(\hat{u})$ and not $\phi(u)$. Then Eq. (3) should be rewritten.

References:

1 Molgedey L, Schuster H: Separation of a mixture of independent signals using time delayed correlations. Phys. Rev. Lett. 1994, 72:3634-3637.

2 A. Belouchrani, K.A. Meraim, J.F. Cardoso, and E. Moulines. A blind source separation technique based on second order statistics. IEEE Trans. on Sig. Proc., 1997.

3 Mueller K, Philips P, Ziehe A: JADE TD: Combining higher-order statistics and temporal information for blind source separation (with noise). Proc Int Workshop on ICA 1999.

4. Clopath C, Longtin A, Gerstner W

An online Hebbian learning rule that performs Independent component analysis

NIPS'07 Proceedings of the 20th International Conference on Neural Information Processing Systems
Pages 321-328, 2007

5. Hyvaerinen A: Complexity pursuit: Separating interesting components from time-series. *Neural Comput* 2001, 13:883-898.

6. An extension of slow feature analysis for nonlinear blind source separation (2014)
H Sprekeler, T Zito, L Wiskott
The Journal of Machine Learning Research 15 (1), 921-947

7. Slowness: An objective for spike timing-dependent plasticity?
H Sprekeler, C Michaelis, L Wiskott
PLoS Computational Biology 3 (6), e112

8. Independent component analysis in spiking neurons
C Savin, P Joshi, J Triesch
PLoS computational biology 6 (4), e1000757

9. Competitive STDP-based spike pattern learning
T Masquelier, R Guyonneau, SJ Thorpe
Neural computation 21 (5), 1259-1276
2009

10. B. Nessler, M. Pfeiffer, L. Buesing, and W. Maass. Bayesian computation emerges in generic cortical microcircuits through spike-timing-dependent plasticity. *PLOS Computational Biology*, 9(4):e1003037, 2013.

11. Inhibitory Synaptic Plasticity-Spike timing dependence and putative network function.
H Sprekeler, TP Vogels, RC Froemke, N Doyon, M Gilson, JS Haas, R Liu, ...
Frontiers in Neural Circuits 7

Ref: NCOMMS-19-12285

Title: Somatodendritic consistency check for temporal feature segmentation

Authors: Toshitake Asabuki and Tomoki Fukai

We thank both reviewers for their constructive criticisms. Below, we have addressed all the concerns raised by the reviewers and included several novel results in the revised manuscript. We have compared performance between the proposed model and other models whenever the comparison is meaningful and also technically possible. We, however, wish to emphasize that the purpose of this study is not to outperform machine learning methods in specific tasks, but to propose a biologically interesting solution (i.e., dendritic computation) to a broad range of important problems in temporal feature analysis without relying on highly task-specialized circuits. In a long run, brain-inspired models might outperform machine-learning methods, or these two approaches might be integrated. At the present stage, however, we do not know the neural code of the brain, and it is not surprising even if a brain-inspired model with biological constraints cannot outperform specialized machine-learning methods in some benchmark.

Here is a brief summary of the major revisions made in the figures. Figures 2C, 4E-G, 7D, 8, and Supplementary Figs. 2, 3 and 5 present results of novel analyses. We have moved the previous Supplementary Figs. 2 and 3 into the present Figs. 3 and 6 to emphasize the wide range of tasks solvable by our model. For clarity, we have divided the previous Fig. 4 into the present Figs. 4 and 5. We have also added several references, most of which were suggested by the reviewers, and slightly modified the title of the manuscript.

Below, slanted fonts indicate the reviewers' comments and normal fonts our reply.

Reviewer #1 (Remarks to the Author):

In this paper the authors claim that a 2 compartment neuron, that minimizes the average surprise between its inputs and its own output, enables learning sequences in an unsupervised manner. They claim that a network of such neurons, connected via mutual inhibition, can perform chunking and BSS. There are a number of concerns about the claims made in the paper and the exact functionality of the network, as outlined below. In addition the authors should discuss and compare their technique against existing techniques (see below). Finally, there are

many small grammatical inconsistencies and stylistic issues. They could benefit from an editing pass by a native English speaker used to scientific prose. These concerns should be addressed before the report is published.

Reply – We thank the reviewer for his/her constructive criticisms. We have carefully considered the reviewer's comments and rewritten our previous claims. We apologize the reviewer for ambiguous wording and grammatical errors. The previous manuscript was indeed checked by a professional editor before submission. Before resubmission, our colleague (a native English speaker) checked the revised manuscript.

The abstract, which should be a standalone piece of text, was unclear in a few places. 1) in the abstract they state, "We show that the same networks of the dendritic...". It is not clear which network the statement is referring to - no networks are mentioned in the prior few sentences. The rest of the abstract is about the power of single neurons, so the relevance of networks is unclear in the abstract. 2) The abstract states "...including the chunking of community structure in temporal sequences" - it's not clear what community they are referring to. It might be clearer to just say "including the chunking of structure in temporal sequences".

Reply – We agree with the reviewer that the previous abstract was unclear on several points. We have corrected the abstract according to the comments by the reviewer. In particular, we have changed the explanation of learning rule as we reformulated the mathematical derivation in response to the comment 2 of the reviewer 2.

Fig 2 - how long did it take the network to create three distinct assemblies? The learning curve is not discussed.

Reply – We have included examples of learning curve in Fig. 2C.

Page 8 - "that human subjects can detect the recurrence of frozen noise patterns embedded in a noisy auditory" - but those studies did not require such extensive learning. In Agus et al., subjects learned the noise after 4 presentations of noise patterns. The learning mechanisms therefore might be significantly different from the method presented here. The authors should comment on this discrepancy in the paper.

Reply – The reviewer pointed out an important difference. We have mentioned this point in the third paragraph of page 5 of the revised manuscript. Whether and how we may reduce the required time of learning remains to be an important open question.

Fig 3 - how is this sequence different from showing three fixed random chunks? What is the significance of the individual letters? The graph structure shows 5 nodes for each chunk, but there are four letters. There are no links between 11 and 15, and 10 and 6 - why is that?

Reply – We used sequences of letters to indicate the potential relevance of the model to language processing. As the reviewer doubted, however, presenting three sequence of four letters to the proposed model is not essentially different from presenting three fixed random chunks. We have included two additional examples in which individual letters have clearer meanings (the third paragraph on page 6). First, we presented the learning of overlapping chunks in which some characters are shared (Fig. 4E-G). In the second example, we have introduced random distractor sequences of arbitrary English letters (a to z) with variable lengths (3 to 7) between the chunks (Supplementary Fig. 3). Here, the network should discriminate the chunks on the basis of individual characters.

In the graph structure (now shown in Fig. 5), we did not introduce any link between 1 and 5, 6 and 10, and 11 and 15 to ensure that each node has the same number of links (i.e., four links).

Fig 3 - what is the significance of the PCA analysis? The significance is not described in the text - there is only one sentence on it.

Reply – We have added the significance of PCA analysis to the caption of Fig. 4D.

Fig 3 - the characters don't overlap. Without overlap the learning is fairly trivial, and almost any system can learn the sequence. What would happen if some of the characters were shared between the chunks, such as "abcd", "efbh", and "iekl"?

Reply – We agree that learning is much easier when chunks contain no overlapping characters. The learning performance of the network for overlapping chunks essentially depends on the timescales of neural dynamics. To show this, we have performed additional simulations for temporal inputs with overlapping chunks. In Fig. 4E-G, we have compared two network models with slow (modified model) or fast (original model) synaptic current, where the former may correspond to the NMDA synaptic current. The

modified model can maintain the recent history of temporal input on much a longer timescale and hence could learn overlapping chunks. The results are explained in the last paragraph of page 6.

Fig 3 - Each "input sequence" is a sequence of random patterns from the input. From Fig 3 (E and F), it looks like the order of the input neuron activity doesn't really matter. Each neuron tends to be active anytime within its preferred "community structure". So why do the authors state the neurons are learning sequence structures? Aren't they just learning to respond to the set of random input patterns that are unique to each chunk? If it truly was learning a temporal sequence, then disrupting the temporal order should cause failures in recognition. This reviewer is unclear as to what the authors mean when they state that their algorithm enables the "self-supervised learning of multiple recurring temporal features" (discussion, page 13).

Reply – The reviewer is correct. The present two-compartment neuron model does not selectively respond to precise sequences, but rather detect the temporal community structure hidden in input. As described in Schapiro et al. (Nat Neurosci 2015), here temporal community is clusters of frequently co-appearing or mutually predicting stimuli in input sequence. This point is mentioned in the first full paragraph of page 7. We have also rewritten the related statements throughout the revised manuscript. However, we speculate that the proposed learning rule enables the detection of precise sequences in more realistic neuron models with multiple dendritic branches. We are examining this speculation and will hopefully report the results in the future. We thank the reviewer to direct our attention to the interesting question.

Fig 4 - As I understand it, there were only two audio files, one for each instrument. The network was trained on a single mixture for 60 repetitions and then tested on that same mixture. This is an extremely restricted experiment, and does not justify the claims made by the authors that their model "achieves BSS from mixtures of correlated signals". Examples of more stringent tests are below:

Reply – We have checked that all the audio files downloaded from the online system correctly work on our Macintosh computer. However, we found the metafiles of some audio files have potential conflicts. Although we are not sure about whether this is the cause of troubles, we have recreated all audio files. We hope that this solves the problems the reviewer experienced. Please note that audio S1 contains one mixed

sound and the original sounds of two instruments in one file. Below, we explain the results of the more stringent tests suggested by the reviewer.

1) Clarinet and bassoon seem to have different average frequencies, and as such it is hard to make any claims that the network is picking up on correlations. Is the network simply picking up on higher vs lower frequencies? What happens if the audio samples have similar frequencies? For example, a clarinet playing two different tunes?

Reply – As exemplified in several figures, the present learning rule learns similar patterns recurring in synaptic inputs, but is not simply picking up on higher vs lower input frequencies (i.e., repeated patterns need not be sinusoidal). According to the suggestion by the reviewer, we have trained the network to discriminate the same instruments playing two different tunes for various sizes of cross-talk noise (please see the Methods for the definition). The novel result is shown in Fig. 8A (middle and right).

2) What happens if you test on a mixture that the network was not trained on exhaustively? This is a more realistic scenario.

Reply – The reviewer raised a quite interesting question. We have studied the question in Fig. 8B. There, the network was first pretrained on each instrument (i.e., for vanishing cross-talk noise) and then tested on a mixture with cross-talk noise of 0.5, up to which the learning performance remains reasonably good (see Fig. 8A). Without additional training on the new mixture, the pre-trained network exhibits much better performance than the untrained network. However, the untrained network rapidly catches up the pre-trained one, and both networks reach the maximal performance after several learning epochs. Thus, pre-training improves the performance for untrained mixtures. As stated in the main text, we used a modified model in all these simulations (please also see our reply below).

3) The authors linearly mixed two sources using a single fixed mixing matrix. This seems unrealistic since one source is weighted twice as much as the other. What happens if other coefficients are used, such as 0.5 and 0.5?

Reply – We have studied how learning performance varies for arbitrary values of cross-talk noise for the mixtures of clarinet and bassoon, clarinet and clarinet, and bassoon and bassoon (Fig. 8A). Because the original model did not perform well for arbitrary

mixtures, we have modified the model by including an additional noise component in the learning rule (equation 17 in the Methods). Performance of the modified model is reasonably good up to cross-talk noise of the magnitude 0.5 to 0.6. Beyond this value, the quality of the decomposed signals gradually drops because the two mixtures given to the network become similar to each other. Note that the two mixtures are identical when the magnitude of cross-talk noise is unity.

Relationship to other work:

1) BSS: On page 11 - The authors state that, “to our knowledge, there are no effective methods for separating mixtures of dependent or correlated signals.” This is a strong claim. In fact, there is a long literature on machine learning methods for BSS. Two examples, focused on single channel source separation, are:

He, P., She, T., Li, W., & Yuan, W. (2018). Single channel blind source separation on the instantaneous mixed signal of multiple dynamic sources. *Mechanical Systems and Signal Processing*, 113, 22–35. <http://doi.org/10.1016/J.YMSSP.2017.04.004>

Hershey, J. R., Chen, Z., Roux, J. Le, & Watanabe, S. (2015). Deep clustering: Discriminative embeddings for segmentation and separation. Retrieved from <http://arxiv.org/abs/1508.04306>

Reply – We are afraid that there was a misunderstanding by the reviewer. In this study, we only consider the cases where two source signals are mutually correlated. From the suggested references about single-channel BSS, however, we speculate that the reviewer interpreted the word “correlation” as autocorrelation of input signals. Because a single-channel BSS is beyond the scope of this paper, we have not compared our method with the other methods. We apologize to the reviewer if our writing was misleading. We have weakened our claim as follows: “However, methods are limited when the mixtures consist of mutually correlated signals.”

We also wish to mention a few words about the comparison suggested by the reviewer. He et al., (2018) does not seem to provide the original code. From our experiences, replication of others’ results is technically very hard without the original codes. In addition, their method maps a single-channel signal onto multi-channel signals prior to source decomposition by FastICA (or its variant). It is therefore unlikely that their method has any advantage in separating the mixtures of mutually correlated signals (please see Fig. 7D). Hershey et al., (2015) provides the source code at github.

However, their method requires pre-training with target signals and hence is not “blind” source decomposition. We feel that the comparison between the “semi-supervised” method and our “unsupervised” method is unfair.

Nevertheless, single-channel BSS remains to be an important future extension of our current model. The mapping strategy described in He et al. (2018) looks useful for such an extension. We thank the reviewer for directing our attention to this interesting approach.

They should perform a more detailed comparison to existing methods, rather than just a naive application of ICA. Python source code for the second reference above is available here: <https://github.com/chaodengusc/DeWave>

Reply – We agree with the reviewer that comparison with a naive ICA is insufficient. As explained later, we have compared our model with temporal ICA, which was suggested by the reviewer #2. Temporal ICA (Second Order Blind Identification or SOBI) is thought to be suitable for separating a mixture of non-independent signals. The new results are included in Fig. 7D, in which we compare learning performance between FastICA, SOBI and the proposed model. When the source signals are mutually independent, all three methods show excellent performance although the two machine learning methods slightly outperform our biology-inspired model (upper panel). When the source signals are mutually correlated, SOBI and our model exhibit significantly better performance than FastICA (bottom). The signals separated by SOBI are presented in Audios S5 and S6.

2) Dendritic learning: They should compare their learning techniques and discuss the following papers which model either dendritic learning and/or sequences in spiking networks:

Legenstein, R., & Maass, W. (2011). Branch-specific plasticity enables self-organization of nonlinear computation in single neurons. The Journal of Neuroscience : The Official Journal of the Society for Neuroscience, 31(30), 10787–802. <http://doi.org/10.1523/JNEUROSCI.5684-10.2011>

Memmesheimer, R. M., Rubin, R., Ölveczky, B., and Sompolinsky, H. (2014). Learning precisely timed spikes. Neuron 82, 925–938. doi: 10.1016/j.neuron.2014.03.026

Hawkins, J., & Ahmad, S. (2016). *Why Neurons Have Thousands of Synapses, a Theory of Sequence Memory in Neocortex*. *Frontiers in Neural Circuits*, 10(23), 1–13. <http://doi.org/10.3389/fncir.2016.00023>

Jahnke, S., Timme, M., and Memmesheimer, R.-M. (2015). *A unified dynamic model for learning, replay, and sharp-wave/ripples*. *J. Neurosci.* 35, 16236–16258. doi: 10.1523/JNEUROSCI.3977-14.2015

Richards, B. A., & Lillicrap, T. P. (2019). *Dendritic solutions to the credit assignment problem*. *Current Opinion in Neurobiology*, 54, 28–36. <http://doi.org/10.1016/J.CONB.2018.08.003>

Sacramento, J., Costa, R. P., Bengio, Y., & Senn, W. (2017). *Dendritic error backpropagation in deep cortical microcircuits*. Retrieved from <http://arxiv.org/abs/1801.00062>

Reply – We have added the subsection “Relationship to previous studies” to the Discussion and discussed the above-listed papers and a few other papers in some details.

“We model this learning process as a self-supervision by single neurons which minimizes time-averaged surprise about spike output for given dendritic activity.” “activity in the dendritic compartment, driven by external inputs, predicts somatic spike responses” - how does this compare to Hawkins & Ahmad where dendritic events are predictions of a neuron’s output? They too use bAP’s to train the dendritic compartment and leads to an unsupervised learning system.

Reply – We have included the following explanation in the second paragraph of page 11: “Hawkins and Ahmad (2016) studied sequence processing by cortical microcircuits in a network model of formal neurons, each of which have apical synapses receiving top-down feedback inputs, proximal synapses receiving feedforward inputs and multiple dendritic segments receiving lateral inputs from nearby neurons. Through spike coincidence detection and dendritic segment-basis Hebbian learning, the network learns sparse activity to predict next spike inputs in sequence. While their model emphasizes the role of dendrites and cortical microcircuit structure in predicting spike sequences, our model demonstrates the ability of single dendritic neurons to learn recurring patterns in temporal input.”

Reviewer #2 (Remarks to the Author):

It is well known that nonlinear Hebbian and generalized STDP algorithms can be used as unsupervised learning rules to perform ICA, sparse coding, spatio-temporal pattern detection, or slow feature analysis (SFA). The paper of Asabuki and Fukai adds another algorithm in this space. Specifically, the novel algorithm combines insights from SFA (ref 42) with insights from the learning rule of Urbanczik and Senn (ref 20).

Reply – We thank the reviewer for his/her overall positive comments to our study and for detailed technical comments on the mathematical formulation of our model. These comments were quite useful for revising the manuscript.

The paper has a nice story line starting with two-compartment neurons and surprise minimization leading to a learning rule similar to the one of Urbanczik-Senn. However, this story line is misleading, because the mathematical derivations that the authors present to derive the learning rule (Eq. 19) do not support this claim. The main steps for the derivations follow the arguments of Urbanczik and Senn, starting from a slightly different objective function. Yet, the objective function DOES NOT give the desired learning rule, because the essential insight enters heuristically (and through the backdoor): the \hat{u} variable is set to unit variance. This aspect is inherited from SFA (as it is correctly acknowledged), but in contrast to SFA where this aspect is part of the objective function and its constraints, it comes in the present paper completely out of the blue.

Reply – We agree with the reviewer that our previous derivation of the learning rule was not mathematically clear enough. As explained below, we have reformulated the learning rule in a more comprehensive fashion in the Methods session. We hope that the novel formalism allows a more straightforward interpretation of the learning rule.

My main concerns are made more precise in the following comments.

MAJOR COMMENTS

1) The algorithm has its merits and may very well be that the new learning rule is stronger than existing unsupervised rules. However, in order to show this, a comparison with other learning rules must be provided.

For ICA (Fig 4), the comparison should be with other TEMPORAL ICA rules (such as Molgedy and Schuster, see refs 1-6 below), and not the ICA rules based on spatial higher-order correlations

cited in the paper. These temporal ICA rules have a known link to asymmetric Hebbian learning (such as STDP).

Reply – We have shown comparison among our method, FastICA and temporal ICA (Second Order Blind Identification or SOBI) in Fig. 7D. When the source signals are mutually independent, all three methods showed excellent performance although FastICA and Temporal ICA methods slightly outperformed our biology-inspired model (upper panel). When the source signals are non-independent (i.e., mutually correlated), Temporal ICA and our model exhibited significantly better performance than FastICA (bottom), as expected. The signals separated by SOBI are presented in Audios S5 and S6. Although Temporal ICA may slightly outperform our model, network implementations of Temporal ICA likely require pre-processing by a delay line, which is not necessary in our network model. Actually, our model does not require highly task-specific network architectures as shown in Fig. 2A. Furthermore, Temporal ICA cannot chunk sequences, which is easy for our model (see Supplementary Fig. 5). We believe that these points show interesting differences between our model and temporal ICA. These points are briefly explained in the second paragraph of page 8.

Also, a comparison with SFA should be done on all three paradigms (Figs 1-4), for example using the SFA version of Sprekeler et al. 2014. (refs 6-8). These SFA rules can be implemented by local Hebbian learning rules and are thus at the same level as the rule in the present manuscript.

Reply – We checked whether the original source codes are publicly available for the slow feature analysis (SFA) version of Sprekeler et al., 2014 or the SFA implemented by STDP (Sprekeler et al., 2007). For both studies, we could not find the source codes. Both models require complicated preprocessing of input data, and implementing their methods in themselves would be independent researches. From our experiences, comparison with published results is quite hard without the original code. Even with the original code, such a comparison can be troublesome if the distributed version contains inaccurate parameter settings (actually, we have recently experienced this trouble in another paper). We have already spent more than three months on the comparisons suggested by the two reviewers: Supplementary Fig. 2 (STDP) and Fig. 7 (Temporal ICA shown below). Due to time limitation, we are unable to perform further comparisons. In addition, SFA needs to be combined with ICA for BSS, meaning that the number of output units in SFA has to coincide with that of input signals. This constraint is biologically unrealistic. We also did not test the model in ref. 9 as it merely gives a

biological implementation of spatial ICA. The type of ICA (FastICA) was tested in Fig. 7D and yielded poor performance.

For spatio-temporal patterns (Figs 1 and 2) the comparison should be with other unsupervised rules that can be used on the same task, such as Masquelier and Thorpe or Nessler et al. (refs 9, 10).

Reply – The results for Masquelier and Thorpe are shown in Supplementary Fig. 2. Again, both Masquelier and Thorpe or Nessler et al. do not provide the original source codes. In particular, the Bayesian network modeled in Nessler et al. is too complicated to correctly reproduce in a short period. However, we were able to create a source code for the method by Masquelier and Thorpe. In doing so, we noticed that the authors implemented their nearest neighbor rule of STDP in a non-standard fashion (this actually caused a delay in our implementation). In their implementation, a postsynaptic neuron has to know which is the nearest neighbor presynaptic spike, past one or future one. This implies that the postsynaptic neuron has to know the times of future presynaptic spikes prior to the occurrence of these events. Without this unrealistic assumption, their model failed to detect spike patterns. Although the authors touched upon this point in their paper, it remains unclear whether the assumption is valid in cortical neurons.

2) The theoretical derivation must be completely rewritten. The current formulation via surprise is completely misleading, because if you were to optimize average surprise WITHOUT the condition on \hat{u} , you would find the constant solution as the best one.

The aim is to arrive at Eq. (19). Question: What is the objective function such that optimization yields Eq. (19)? Somehow the constraint on \hat{u} must become part of the derivation - otherwise the derivation remains a copy of the theory of Urbanczik and Senn. My guess is that similar to variational autoencoders (I mean the encoding step, not the decoding step) you could require that the distribution of \hat{u} has unit variance and zero mean, for example. Can you define a KL divergence so that v will approach \hat{u} ? Then optimize this and arrive directly at Eq. (19)? It might not be easy, but somehow the constraints on \hat{u} must be part of the formalism.

Reply – We are very much grateful to the reviewer for the valuable suggestion. Indeed, we were also a little uncomfortable with the previous interpretation of the learning rule. According to the suggestion, we have reformulated our learning rule in terms of a KL divergence (see the Methods). The novel formalism does not change the mathematical

expression of learning rule, but it provides a clearer insight into the computational principles behind the rule and their biological implementations. In particular, we believe that the new formalism clarifies the differences between our model and that of Urbanczik and Senn (we will explain these points below).

3) *It is not even clear that you need a two-compartment neuron.*

You could also consider \hat{u} as some low-pass filtered version of v^ that is represented in the Ca-dynamics or similar. Define three online low-pass variables for v^* , $(v^*)^2$, and a normalized combination for these (your \hat{u}).*

Reply – As the reviewer pointed out, whether the soma and dendrite are separately required for the proposed learning remained unclear in the previous manuscript. We believe that this point becomes clearer in the novel formalism, in which a two-compartment neuron minimizes a KL divergence between two distinct probability densities, one representing the statistics of input activity and the other representing the statistics of spiking output. Although the underlying biological mechanisms still need to be clarified in detail, associating the two probability densities with separate dynamical units, i.e., the soma and dendrite, does not seem to be an unrealistic choice.

4) *For an additional point of view of your novel learning rule, you may consider the following analogy.*

*In a general BCM-type learning rule, the weight updates are proportional to $(d/dt)w_{\{ij\}} = \eta [\Psi(v^*_i)[\phi(v^*_i) - \theta(v^*_{bar_i})] e_j - \gamma w_{\{ij\}}$ where $v^*_{bar_i}$ is some running average of the postsynaptic activity v^*_i . See Bienenstock et al. 1982.*

*I can rewrite your Eq. (19) similarly as $(d/dt)w_{\{ij\}} = \eta [\theta(v^*_{bar_i}) - \phi(v^*_i)] e_j - \gamma w_{\{ij\}}$ where $v^*_{bar_i}$ is the normalized running average of the postsynaptic activity v^*_i (your \hat{u}).*

What are the main differences?

*(i) There is a sign difference. In BCM $\theta(v^*_{bar})$ is an UNSTABLE fixed point. This instability is necessary to create sensitivity to fluctuations that the learning rule needs to enhance to form receptive fields or find independent components. If you switch the sign, this turns into a stable fixed point. Normally this would be a boring fixed point because no receptive fields or*

independent components can be learned, the neuron just normalises its rate to a desired mean. However:

(ii) the second difference is that v^ in YOUR case is normalised so as to have a standard deviation of one (like in SFA). So the learning rule cannot converge to a fixed point, but must adapt the weights so that v^* tries to follow the normalized fluctuations of v^* . Thus the sensitivity to fluctuations comes no longer from the learning rule we see in your Eq. (19), but from the definition of v^* as a NORMALIZED running average.*

May be these analogies help to rewrite the paper.

Reply – We thank the reviewer for these comments. We consider that the reviewer's thought about the relationship between our learning rule and BCM theory is correct. The sign difference between the two learning rules explains why we have to avoid a trivial solution of our stable learning rule by imposing additional conditions on the somatic responses (as we will explain later). We briefly explained these points below equation (16) in the Methods.

5) The story-line of the paper has to be rewritten from scratch. In my opinion the learning rule has nothing to do with surprise, but a lot with making the v -distribution close to the u -hat distribution which is BY DEFINITION normalized. See the points 2-4 above.

Reply – As we explained in our reply to comment 4, we have reformulated the learning rule without relying on surprise. We have rewritten the Introduction according to the novel formalism of the learning rule.

6) Unsupervised versus supervised; generative model versus recognition model.

You seem to imply that Urbanczik and Senn only use a supervised paradigm. However, they already studied an unsupervised paradigm with lateral interactions (but not the single neuron). So, explain better please what is different in your case.

Reply – The reviewer raised an important question. As he/she mentioned, Urbanczik and Senn showed an example of self-organization in a network model of their neuron model (i.e., topographic map formation). However, they considered a special case where the network architecture had a topographic map close to the desirable one prior to learning. In addition to this, distant-dependent recurrent synapses provided effective teaching signals to the somatic components. Therefore, in their model unsupervised

learning was not specified by the learning rule but by the network architecture. Indeed, the authors commented on this point when they explained Figure 3 in their paper.

In contrast, our neuron model requires neither explicit teaching signal nor preconfigured network structure. As demonstrated in Fig. 1, unsupervised learning occurs within a single neuron. Importantly, our model can perform the present learning tasks only if the somatic activity undergoes the history-dependent modulations of somatic response function (eqs. (4)-(7)). We speculate that the recent history of somatic activity provides an effective teaching signal for learning temporal features. It is noted that the model by Urbanczik and Senn, which does not have the above modifications, also cannot perform the present unsupervised learning tasks.

These important points were unclear in the previous manuscript. We have explained these important points in the beginning of the subsection “Relationship to previous models” (page 10). We have also explained that the somatic response modifications may be achieved in cortical neurons by local inhibitory circuits (Murayama et al., Nature 2009) or plasticity of intrinsic excitability (Titley et al., Neuron 2017).

Moreover you seem to imply that sparse coding or ICA or variational autoencoders are different because they aim at a reconstruction of the original signal. However, the Dayan-Abbott textbook (see Chapter 10, and table in appendix 10.6) explains that the generative model view ('reconstruction') and the recognition view ('decoding') and the Hebbian learning rule view (online local algorithm) are just three different, but equivalent, views of the SAME learning algorithm. This parallelism has been rediscovered and discussed by many people over the last 20 years. For example, one can apply the Olshausen learning rule in order to learn a sparse representation WITHOUT the framework of stimulus reconstruction. And follow-up work from Friedrich Sommer in the Olshausen group has shown that one can also just see it as an unsupervised learning rule, not even bothering about the coding.

Reply – I apologize the reviewer if our previous descriptions were misleading. It is true that a generative model, a reconstruction model and an online local algorithm can describe an equivalent probability model, and the corresponding learning rules also represent essentially the same learning rule. Such a parallelism will hold if the different algorithms capture essentially the same amount of information on the external environment. However, there are also a lot of cases in which some learning algorithm gets more information than others. It is in this sense that our model may differ from the other algorithms for information compression. For example, our algorithm is sensitive to the community structure of temporal input but, at least in its current implementation,

not sensitive to the serial order of input data. This limits the ability of the model to predict a future input from a previous data sequence. In this sense, the model loses a certain amount of information about input streams. Nevertheless, a family of the network models we proposed perform a wide range of temporal feature analyses including chunking and BSS. We believe that these findings are important and that our model achieves a significant insight into the basic circuit functions of cortical information processing. To avoid confusions, we have rewritten the misleading paragraph in the subsection “Comparison with other computational principles” of the Discussion.

MINOR COMMENTS:

p.5, lines 20-25. As you know, real neurons are either excitatory or inhibitory. Unclear whether you think of this learning rule like a rule for the inhibitory synapses (see review of Sprekeler et al. 2013) or of the excitatory neurons onto inhibitory neurons (as your reference seems to suggest).

Reply – Here, inhibition between pyramidal cells should be weakened to form feature-detecting cell assemblies when both neurons respond to the same temporal feature. As suggested in the manuscript, this change can occur when excitatory synapses onto inhibitory neurons follow the iSTDP rule shown in the paper. However, the iSTDP rule implemented at inhibitory synapses should also induce the required changes in the inhibitory circuit. We have explained these points in the first paragraph of page 5, citing the related reference.

p 15. Eq. 2. I understand from the main text that you use spiking neurons. But then the last term in Eq. 2 should be a sum of the spike trains of other neurons, and not over their firing rates.

Reply - We apologize to the reviewer for our ambiguous explanation. Although we generated Poisson spike trains of output neurons for simulating STDP at inhibitory synapses, we calculated lateral inhibition by using the firing rates. We have explained these points below equation 2 in the revised manuscript.

p15. Eq. 3. I understand that spikes are in the end drawn from $\phi(u)$ and not $\phi(u)$. Then Eq. (3) should be rewritten.

Reply - We have entirely reformulated the learning rule including equation 3. In the revised formulation, the somatic membrane potential u determines the rate of somatic

spikes through the rate function given in equation 3. Now, the slope and threshold of rate function are modified according to the time-averaged membrane dynamics. As explained below equation (7), these manipulations induce similar effects to the previous standardization by what on the membrane dynamics.

References:

1 Molgedey L, Schuster H: *Separation of a mixture of independent signals using time delayed correlations.* *Phys. Rev. Lett.* 1994, 72:3634-3637.4.

2 A. Belouchrani, KA. Meraim, JF. Cardoso, and E. Moulines. *A blind source separation technique based on second order statistics.* *IEEE Trans. on Sig. Proc.*, 1997.

3 Mueller K, Philips P, Ziehe A: *JADE TD: Combining higher-order statistics and temporal information for blind source separation (with noise).* *Proc Int Workshop on ICA 1999.*

4 Clopath C, Longtin A, Gerstner W. *An online Hebbian learning rule that performs Independent component analysis, NIPS'07 Proceedings of the 20th International Conference on Neural Information Processing Systems Pages 321-328, 2007*

5. Hyvaerinen A: *Complexity pursuit: Separating interesting components from time-series.* *Neural Comput 2001, 13:883-898.*

6. *An extension of slow feature analysis for nonlinear blind source separation (2014)*
H Sprekeler, T Zito, L Wiskott, *The Journal of Machine Learning Research 15 (1), 921-947*

7. *Slowness: An objective for spike timing-dependent plasticity?*
H Sprekeler, C Michaelis, L Wiskott, *PLoS Computational Biology 3 (6), e112*

8. *Independent component analysis in spiking neurons*
C Savin, P Joshi, J Triesch, *PLoS computational biology 6 (4), e1000757*

9. *Competitive STDP-based spike pattern learning*
T Masquelier, R Guyonneau, SJ Thorpe, *Neural computation 21 (5), 1259-1276, 2009*

10. B. Nessler, M. Pfeiffer, L. Buesing, and W. Maass. *Bayesian computation emerges in generic cortical microcircuits through spike-timing-dependent plasticity*. *PLOS Computational Biology*, 9(4):e1003037, 2013.

11. *Inhibitory Synaptic Plasticity-Spike timing dependence and putative network function*.

H Sprekeler, TP Vogels, RC Froemke, N Doyon, M Gilson, JS Haas, R Liu, ...
Frontiers in Neural Circuits 7

Reviewers' Comments:

Reviewer #1:

Remarks to the Author:

The authors have addressed my concerns. The abstract, introduction, and results are significantly improved and the overall claims of the paper are clearer. There are not many published serious models based on multi-compartment dendritic learning that achieve functional results, and this paper will help add to the literature. I have no further comments, and approve the article for publication.

Reviewer #2:

Remarks to the Author:

This new version of the paper is significantly better than the previous one. In particular, the new mathematical derivation is now better linked to the reformulated claims.

Also some additional comparisons with other algorithms have been done which I appreciate.

However, I still have major concerns, mainly concerning the formulation of the main message, the context, and the comparisons with existing algorithms where additional work needs to be done. Once these changes are implemented, I recommend acceptance of the paper.

1) Formulation of the main message in the introduction.

1.1. The introduction should end in line 67 with a summary of the main idea of the algorithm.

Moreover, in the present manuscript some of the most important references are spread out over various places in the paper and methods, but should be collected at the end of the introduction.

Here my PROPOSITION for three new sentences at the end of the abstract:

Our algorithm combines ideas of the two-compartment learning rule of Urbanczik and Senn (ref 21) with insights from SFA (ref 56) and ICA based on temporal correlations (ref 38). A central feature of our learning rule is that synaptic weights on the dendrite are changed such that the somatic membrane potential fluctuates with unit variance around a target value μ . Our formulation is inspired by observations that neuronal adaptation shifts the neuron always toward a regime of efficient information transmission (new references 1 -4 below).

1.2. Introduction, Line 52f. Replace the sentence:

Furthermore, our model predicts the gain and threshold of somatic responses should be modulated in an activity-history-dependent manner.

By the following formulation. PROPOSITION

Importantly, our learning rule exploits the fact that neuronal adaptation is able to maintain somatic membrane potential in a regime where spiking has high information content (new references 1 -4 below). Therefore the gain and threshold of the somatic transfer function in our model are adapted in

a history-dependent manner.

2) References and context.

The authors should state more clearly that temporal ICA, SFA, and nonlinear Hebbian learning are closely related to each other and that their algorithm falls in this family of algorithms. The authors acknowledge this somehow in the response to the reviewers, but the authors have not yet fully responded to this point in the main text. At present the reader of the paper is not informed about this. A good location for placement in the context would be at the beginning of the discussion.

PROPOSITION. A possible formulation could be:

It is well known that nonlinear Hebbian and generalized STDP algorithms can be used as unsupervised learning rules to perform receptive field development, ICA, sparse coding, spatio-temporal pattern detection, or SFA (with new references 5-8 below). Our new algorithm belongs to the same family of methods and is applied to some classic problems of receptive field development and ICA as well as to the additional problem of 'chunking' as an example of a task with rather specific temporal structure that has traditionally been solved with more specialized algorithms (your citations).

3. Quantitative comparison with existing algorithms.

I acknowledge that the authors made a few big steps in this direction in the new version of the paper. However, the manuscript still has short-comings that do not allow a fair comparison between competitors.

The MRIL algorithm has at least 4 free parameters (the main ones being θ_0 , ϕ_0 and t_0 , but also τ_s) which have all been varied by a factor of three to ten between one task and the next. A fair comparison with competitors requires that for the other algorithms you also tune parameters between one task and the next. It was not clear to me whether this was done.

For example, you claim in the text that the SOBI algorithm did not chunk the English characters (line 231). Did you try to optimize the time constant (delay τ) of that algorithm?

In case there is no explicit time delay constant in the version of the SOBI algorithm that you use, try the following. You low-pass the raw input with 50ms time constant (just as you do in your MRIL algo), and then you resample in discrete time (e.g. with 10ms steps) before you apply SOBI (with standard parameters).

Also, why not try SFA on this task? SFA has also a few parameters that you can optimize separately for each task.

In my opinion, SFA is the algorithm that is closest to MRIL. As opposed to what you write in the response to referees it is not correct that the algorithm is not available in downloadable form.

A simple google search gave pointers to the scholarpedia article on SFA, in which I found links to a matlab SFA toolkit (search for sfa-tk if the link in the page does not work)

<https://pberkes.github.io/software/sfa-tk/index.html>

as well as a modular toolkit in python that mentions in the welcome page that SFA is part of the package:

<http://mdp-toolkit.sourceforge.net/>

I insist on the comparison simply because you write at several places in the paper, and even in the abstract, that '... methods applicable to these temporal feature analyses were previously unknown'. However, I do believe that SFA is a natural candidate method for 'temporal feature analysis'. If you do not want to compare with competing algorithms such as SFA, then you have to remove this statement in the abstract and choose formulations/claims more carefully throughout the text.

I understand your argument in the response to reviewers that the main merits of MRIL are on the biological side and not necessarily on the application side. In fact, I actually agree with you on that point. There is nothing wrong in presenting an algorithm that is comparable to SFA (be it as strong, nearly as strong, stronger or somewhat weaker), if this new algo has a much better biological interpretability. I am completely with you on this point - but then you will have to formulate the claims more carefully.

I suggest that the authors spend 4 weeks on trying to make the SFA algorithm work (time well invested for an article in Nature Communications), but in the end, it is the authors decision whether they rather want to lower the claims.

A first simple solution could also be to remove this sentence from the abstract. And remove similar claims in the main text.

4. Minor points.

- Abstract, line 18

You don't need to follow this specific suggestion, because it is a matter of taste, but I would propose to replace the sentence

By analogy with this effect, we model a self-supervising process by single neurons to minimize differences in the probabilistic responses between somatic and dendritic activities.

(I found this hard to understand - what are probabilistic responses at the dendrite? Dendrites do not spike, but the KL is over the spike distribution, not the input distribution.)

by the following formulation PROPOSITION:

By analogy with this effect, we model a self-supervised synaptic plasticity process that increases the similarity between the dendritic membrane potential and the somatic one where the somatic membrane potential distribution is normalized by a running average to fixed mean and unit variance.

(it's a bit longer, but much more concrete.)

- line 72-75 grammar/wording. Turn into two separate sentences

- abstract, as well as line 98, 120 and other places

'Dendritic neuron'. Unclear. Better: two-compartment neuron
OR neurons with a dendrite

- line 105:
three fixed temporal patterns
--: three fixed temporal patterns of 20ms each

- line 121 iSTDP rule
'This rule weakens inhibition between two neurons which respond to the same feature'

I actually conjecture that it is the exact opposite that you have implemented: you increase inhibition between two neurons which respond to the same feature.

(I guess the confusion comes from the fact that you have negative weights: if the amplitude of negative weights gets bigger, then you increase inhibition, and the weight value goes down = more negative).

If my interpretation is correct, then your interaction algo is very close to both the Vogels rule and Foldiak's decorrelation principle and both should be cited (new references 10 and 11 below).

If my interpretation is wrong, then you need to explain why you do the opposite of what the field of computational neuroscience is doing - Foldiak and Vogels are by no means the only ones to try to decorrelate neurons (see also Zylberberg, or Fritz Sommer, and have a look at those papers that cite Foldiak).

In the reply to the reviewers you insist on your interpretation ('This rule WEAKENS inhibition between two neurons when both of them respond to the same feature').

But just to be sure, please check again in your code. Intuitively, would have implemented the plasticity rule with the opposite sign so that ('This rule INCREASES inhibition between two neurons which respond to the same feature').

- line 143. Regularization by noise.
Note that you can also think of your tests with modified input as data augmentation (have a look at any standard machine learning book).

- line 166-168 ... know the time of future presynaptic spikes.

This statement about STDP is clearly wrong and should be removed. The integral formulation in the form of an STDP window may suggest such an interpretation, but it is well known since the work of Song and Abbott or Kistler and van Hemmen that pair-based STDP can be implemented by local variables running forward in time, without the need to look into the future. See for example the text book 'Neuronal Dynamics', box at the end of section 19.2.2., equations 19.12- 19.14. Similar statements hold also for STDP with nearest-neighbor interactions (see a paper of Abigail Morrison in Biological Cybernetics)

I take it from your response to the referee that you might actually refer to an additional, more subtle point in the implementation of Masquelier of which I am not aware. This is absolutely possible, but then

the sentence in line 166-168 is still not explicit enough to clarify this point and the easiest solution would be to remove it.

- line 190 - 192, grammar/wording/logic

- line 264/265 ... and somatic output.

BETTER

... and somatic output in the presence of neuronal adaptation.

- line 302

inhibitory circuits or intrinsic excitability.

ALSO MENTION

... or neuronal adaptation

- line 368/369

The reader may be confused at this stage by the superscript 'dend' on the distribution. You are talking at this stage only about a somatic distribution, and then this somatic distribution is rewritten as a normalized distribution.

I suggest to drop the superscript 'dend' and instead note the (normalized) distribution as $\hat{\phi}$ (with a ^ for hat).

- line 386-387 TURN SENTENCE AROUND, BECAUSE THE LOGIC IS NOT CLEAR

OLD:

We assumed that an attenuated version ν^* of dendritic potential well describes the somatic membrane potential with the degree of attenuation α Explicitly representing ...

NEW PROPOSITION:

In a stationary state, the somatic membrane potential u of a two-compartment model can be written as an attenuated version ν^* of the dendritic membrane potential with an attenuation factor $\alpha =$... (cite 21). Even though we work with time-dependent stimuli, we compare in our model at each point in time the attenuated dendritic membrane potential ν^* with the somatic membrane potential u .

The comparison is not down directly on the level of the membrane potential but on the level of the two Poissonian spike distributions with rates

$\phi_i^{\text{som}}(u)$ and $\hat{\phi}(\nu^*)$, respectively, that would be generated if both soma and dendrite were able to emit spikes independently. Explicitly representing ...

line 396

We search for ...

NEW PROPOSITION ...

Similar to reference (21), we search for ...

line 435

cite both Foldiak and Vogels.

line 473 The algorithm should be made publicly available on some web page or repository at the moment when the paper is accepted.

References:

Adaptation and coding:

1. Efficiency and ambiguity in an adaptive neural code
AL Fairhall, GD Lewen, W Bialek, RRR van Steveninck
Nature 412 (6849), 787
2. Shifts in coding properties and maintenance of information transmission during adaptation in barrel cortex
M Maravall, RS Petersen, AL Fairhall, E Arabzadeh, ME Diamond
PLoS biology 5 (2), e19
3. S. Mensi, O. Hagens, W. Gerstner, and C. Pozzorini (2016)
Enhanced Sensitivity to Rapid Input Fluctuations by Nonlinear Threshold Dynamics in Neocortical Pyramidal Neurons
PLoS Comput Biol 12: e1004761. doi:10.1371/journal.pcbi.1004761
4. C. Pozzorini, R. Naud, S. Mensi, and W. Gerstner (2013)
Temporal whitening by power-law adaptation in neocortical neurons
Nature Neuroscience 16:942 - 948

LINK OF ICA and SFA and STDP

5. Clopath C, Longtin A, Gerstner W
An online Hebbian learning rule that performs Independent component analysis
NIPS'07 Proceedings of the 20th International Conference on Neural Information Processing Systems
Pages 321-328, 2007
6. An extension of slow feature analysis for nonlinear blind source separation (2014)
H Sprekeler, T Zito, L Wiskott
The Journal of Machine Learning Research 15 (1), 921-947
7. Slowness: An objective for spike timing-dependent plasticity? (2007)
H Sprekeler, C Michaelis, L Wiskott
PLoS Computational Biology 3 (6), e112
8. Independent component analysis in spiking neurons (2010)
C Savin, P Joshi, J Triesch
PLoS computational biology 6 (4), e1000757

INHIBITORY PLASTICITY

10. TP Vogels, H Sprekeler, F Zenke, C Clopath, W Gerstner
Inhibitory Plasticity Balances Excitation and Inhibition in Sensory Pathways and Memory Networks
Science 334 (6062), 1569-1573 (2011)
11. P. Földiák (1990) Forming sparse representations by local anti-Hebbian learning

Biol. Cybern. 64, 165-170 (1990)

I believe that after implementing these changes (and ideally with some additional comparisons) the manuscript is going to be a very nice paper for publication in Nature Communications that will find a broad readership

SIGNED

Wulfram Gerstner

Ref: NCOMMS-19-12285A

Title: Somatodendritic consistency check for temporal feature segmentation

Authors: Toshitake Asabuki and Tomoki Fukai

We thank both reviewers for their positive comments on our revised manuscript. According to the criticisms raised by the 2nd reviewer, we have performed additional simulations for comparison purposes and shown the novel results in Supplementary Figs. 2 and 6. Consequently, the number of supplementary figures has been increased by one and now is six. We have also included brief explanations of SOBI and SFA in the Methods. These pieces of information were missing in the previous version. We have also corrected errors in some equations and the values of parameters.

Below, we summarize the changes made in the manuscript in response to the reviewer's comments. Slanted fonts show the reviewers' comments and bold fonts our reply. We hope that the revised manuscripts meet all the concerns raised by the reviewer.

Reviewer #2 (Remarks to the Author):

This new version of the paper is significantly better than the previous one. In particular, the new mathematical derivation is now better linked to the reformulated claims. Also some additional comparisons with other algorithms have been done which I appreciate.

However, I still have major concerns, mainly concerning the formulation of the main message, the context, and the comparisons with existing algorithms where additional work needs to be done. Once these changes are implemented, I recommend acceptance of the paper.

Reply – We thank prof. Gerstner for his constructive criticisms, which we felt were useful for further strengthening our manuscript.

1) Formulation of the main message in the introduction.

1.1. The introduction should end in line 67

with a summary of the main idea of the algorithm.

Moreover, in the present manuscript some of the most important references are spread out over various places in the paper and methods, but should be collected at the end of the introduction.

Here my PROPOSITION for three new sentences at the end of the abstract:

Our algorithm combines ideas of the two-compartment learning rule of Urbanczik and Senn (ref 21) with insights from SFA (ref 56) and ICA based on temporal correlations (ref 38). A central feature of our learning rule is that synaptic weights on the dendrite are changed such that the somatic membrane potential fluctuates with unit variance around a target value μ . Our formulation is inspired by observations that neuronal adaptation shifts the neuron always toward a regime of efficient information transmission (new references 1 -4 below).

Reply – We believe that the reviewer proposed to add the three sentences at the end of the introduction, not the abstract. After minor modifications, we have added these sentences to the revised manuscript at lines 69-75. The suggested new references have been cited.

1.2. Introduction, Line 52f. Replace the sentence:

Furthermore, our model predicts the gain and threshold of somatic responses should be modulated in an activity-history-dependent manner.

By the following formulation. PROPOSITION

Importantly, our learning rule exploits the fact that neuronal adaptation is able to maintain somatic membrane potential in a regime where spiking has high information content (new references 1 -4 below). Therefore the gain and threshold of the somatic transfer function in our model are adapted in a history-dependent manner.

Reply – We agree with the reviewer on the function importance of adaptation of somatic transfer function in our model. We have added the above sentences to lines 52-55.

2) References and context.

The authors should state more clearly that temporal ICA, SFA, and nonlinear Hebbian learning

are closely related to each other and that their algorithm falls in this family of algorithms. The authors acknowledge this somehow in the response to the reviewers, but the authors have not yet fully responded to this point in the main text. At present the reader of the paper is not informed about this. A good location for placement in the context would be at the beginning of the discussion.

PROPOSITION. A possible formulation could be:

It is well known that nonlinear Hebbian and generalized STDP algorithms can be used as unsupervised learning rules to perform receptive field development, ICA, sparse coding, spatio-temporal pattern detection, or SFA (with new references 5-8 below). Our new algorithm belongs to the same family of methods and is applied to some classic problems of receptive field development and ICA as well as to the additional problem of 'chunking' as an example of a task with rather specific temporal structure that has traditionally been solved with more specialized algorithms (your citations).

Reply – We have stated the above sentences at the beginning of the discussion after minor corrections.

3. Quantitative comparison with existing algorithms.

I acknowledge that the authors made a few big steps in this direction in the new version of the paper. However, the manuscript still has short-comings that do not allow a fair comparison between competitors.

The MRIL algorithm has at least 4 free parameters (the main ones being θ_0 , ϕ_0 and t_0 , but also τ_s) which have all been varied by a factor of three to ten between one task and the next. A fair comparison with competitors requires that for the other algorithms you also tune parameters between one task and the next. It was not clear to me whether this was done.

For example, you claim in the text that the SOBI algorithm did not chunk the English characters (line 231). Did you try to optimize the time constant (delay τ) of that algorithm?

In case there is no explicit time delay constant in the version of the SOBI algorithm that you use, try the following. You low-pass the raw input with 50ms time constant (just as you do in your MRIL algo), and then you resample in discrete time (e.g. with 10ms steps) before you apply SOBI (with standard parameters).

Reply – We thank the reviewer for pointing out the important factors missed in our comparison. We have examined what the reviewer mentioned in the above paragraph with the parameter values suggested by the reviewer. A typical result from our repeated simulations for SOBI is shown in Supplementary Figure 6A. In our simulations, SOBI could not generate highly chunk-selective responses. Rather, most of the units responded to all three chunks in SOBI. We have conducted similar analyses for low-pass filtered versions of the input by using different time constants for coarse graining (15, 30 and 50 ms) or the bin width (1 ms or 10 ms), but the essential results remained unchanged. Based on these results, we have rewritten the related texts (ll. 248-254).

Also, why not try SFA on this task? SFA has also a few parameters that you can optimize separately for each task. In my opinion, SFA is the algorithm that is closest to MRIL. As opposed to what you write in the response to referees it is not correct that the algorithm is not available in downloadable form.

A simple google search gave pointers to the scholarpedia article on SFA, in which I found links to a matlab SFA toolkit (search for sfa-tk if the link in the page does not work)

<https://pberkes.github.io/software/sfa-tk/index.html>

as well as a modular toolkit in python that mentions in the welcome page that SFA is part of the package:

<http://mdp-toolkit.sourceforge.net/>

Reply – We thank the reviewer for the information about toolkits. We have tried the python SFA toolkit. In all our trials, the toolkit never stopped with a stable solution when input sequences involved chunks. As the toolkit worked properly for sequences without chunks, we consider the toolkit itself worked correctly in our simulations. We speculate that SFA is in principle inadequate for chunk detection. SFA attempts to minimize the average changes of signal in local temporal domains. However, detecting a whole chunk and detecting an arbitrary single character cost equally in the objective function of SFA. Due to this fact, the minimization algorithm of SFA presumably has too many solutions to chunking. We have mentioned these results and interpretation at ll. 254-259.

I insist on the comparison simply because you write at several places in the paper, and even in

the abstract, that '... methods applicable to these temporal feature analyses were previously unknown'. However, I do believe that SFA is a natural candidate method for 'temporal feature analysis'. If you do not want to compare with competing algorithms such as SFA, then you have to remove this statement in the abstract and choose formulations/claims more carefully throughout the text.

I understand your argument in the response to reviewers that the main merits of MRIL are on the biological side and not necessarily on the application side. In fact, I actually agree with you on that point. There is nothing wrong in presenting an algorithm that is comparable to SFA (be it as strong, nearly as strong, stronger or somewhat weaker), if this new algo has a much better biological interpretability. I am completely with you on this point - but then you will have to formulate the claims more carefully.

I suggest that the authors spend 4 weeks on trying to make the SFA algorithm work (time well invested for an article in Nature Communications), but in the end, it is the authors decision whether they rather want to lower the claims.

A first simple solution could also be to remove this sentence from the abstract. And remove similar claims in the main text.

Reply – Indeed, we appreciate the reviewer’s suggestions on our previous comparison and decided to spend more time on the comparison (However, this comparison took longer than we expected because the algorithm of SFA did not work properly in the beginning). We believe that the novel results have made the differences between our method and SFA/ICA clearer than before. We hope that the revised manuscript successfully meets the criticisms by the reviewer.

4. Minor points.

- Abstract, line 18

You don't need to follow this specific suggestion, because it is a matter of taste, but I would propose to replace the sentence

By analogy with this effect, we model a self-supervising process by single neurons to minimize differences in the probabilistic responses between somatic and dendritic activities.

(I found this hard to understand - what are probabilistic responses at the dendrite? Dendrites do not spike, but the KL is over the spike distribution, not the input distribution.)

by the following formulation PROPOSITION:

By analogy with this effect, we model a self-supervised synaptic plasticity process that increases the similarity between the dendritic membrane potential and the somatic one where the somatic membrane potential distribution is normalized by a running average to fixed mean and unit variance.

(it's a bit longer, but much more concrete.)

Reply – We thank the reviewer for the suggestions. We agree that the “probabilistic responses of dendrite” sounds ambiguous. According to the suggestion, we replaced the sentence as shown below. The revised sentence may still sound somewhat abstract. However, we had to shorten the sentence due to the length limit of the journal (the length of the revised abstract coincides with the limit of 150 words).

“By analogy with this effect, we model a self-supervising process that increases the similarity between dendritic and somatic activities where the somatic activity is normalized by a running average.”

- line 72-75 grammar/wording. Turn into two separate sentences

- abstract, as well as line 98, 120 and other places

'Dendritic neuron'. Unclear. Better: two-compartment neuron

OR neurons with a dendrite

Reply – We have replaced “dendritic neurons” in the abstract as follows: dendritic neuron -> two-compartment neuron; networks of dendritic neurons -> neural networks with dendrites. In other places, we have used either “neurons with dendrites” or “two-compartment neuron” depending on the context of arguments.

- line 105:

three fixed temporal patterns

--: three fixed temporal patterns of 20ms each

Reply – We have modified the text according to the suggestion by the reviewer.

- line 121 iSTDP rule

'This rule weakens inhibition between two neurons which respond to the same feature'

I actually conjecture that it is the exact opposite that you have implemented: you increase inhibition between two neurons which respond to the same feature.

(I guess the confusion comes from the fact that you have negative weights: if the amplitude of negative weights gets bigger, then you increase inhibition, and the weight value goes down = more negative).

Reply – Negative peak around a zero time difference ($Dt=0$) means that changes in inhibitory weights take large negative value in our iSTDP rule. Therefore, inhibitory weights are actually weakened when pre- and post-synaptic neurons fire synchronously. We have clarified these points at lines 128-135 in the revised manuscript. In the simulations shown in Fig. 2, inhibitory weights were bounded in the positive regime. This constraint was previously explained in the Methods section, but we have added a brief explanation to the above mentioned location of the main text.

If my interpretation is correct, then your interaction algo is very close to both the Vogels rule and Foldiaks decorrelation principle and both should be cited (new references 10 and 11 below).

If my interpretation is wrong, then you need to explain why you do the opposite of what the field of computational neuroscience is doing - Foldiak and Vogels are by no means the only ones to try to decorrelate neurons (see also Zylberberg, or Fritz Sommer, and have a look at those papers that cite Foldiak).

Reply – As we mentioned above, our iSTDP is actually different from that of Foldiak and Vogels. For comparison, we have performed novel simulations using the conventional STDP rule and cited Foldiak and Vogels as well as Zylberberg et al. with the novel results. These results are explained in a separate paragraph at ll. 144-153 (please also see below).

In the reply to the reviewers you insist on your interpretation ('This rule WEAKENS inhibition between two neurons when both of them respond to the same feature'). But just to be sure, please check again in your code. Intuitively, would have implemented the plasticity rule with the opposite sign so that ('This rule INCREASES inhibition between two neurons which respond to the same feature').

Reply – We have checked our code to confirm that our rule actually weakens inhibitory synapses when two neurons responded to the same features. As mentioned above, we have implemented the rule by Foldiak and Vogels to our network model, that is, we strengthened inhibitory synapses when presynaptic and postsynaptic neurons fired synchronously. The same temporal inputs as used in Fig. 2 were also used in the novel simulations. The results are shown in Supplementary Fig. 2.

As expected, the conventional iSTDP rule (Supplementary Fig. 2A) generated variety of complex response patterns of which only a small portion expressed chunk-specific responses (Supplementary Fig. 2B). Some neurons responded to more than one chunk (e.g., neurons 1 and 10) and other neurons to both chunks and random inputs almost arbitrarily (e.g., neuron 5). Inhibitory weight matrix also showed no obvious cell-assembly structure (Supplementary Fig. 2C). Therefore, our iSTDP rule is thought to be more suitable than the conventional one at least for the present chunk-detection task.

- line 143. Regularization by noise.

Note that you can also think of your tests with modified input as data augmentation (have a look at any standard machine learning book).

Reply – We thank the reviewer for the comment. We have briefly mentioned the viewpoint at lines 161-162.

- line 166-168 ... know the time of future presynaptic spikes.

This statement about STDP is clearly wrong and should be removed. The integral formulation in the form of an STDP window may suggest such an interpretation, but it is well known since the work of Song and Abbott or Kistler and van Hemmen that pair-based STDP can be implemented by local variables running forward in time, without the need to look into the future. See for example the text book 'Neuronal Dynamics', box at the end of section 19.2.2., equations 19.12-19.14. Similar statements hold also for STDP with nearest-neighbor interactions (see a paper of Abigail Morrison in Biological Cybernetics)

I take it from your response to the referee that you might actually refer to an additional, more subtle point in the implementation of Masquelier of which I am not aware. This is absolutely possible, but then the sentence in line 166-168 is still not explicit enough to clarify this point and the easiest solution would be to remove it.

Reply – According to the comments, we have eliminated the sentence. We, however, clarify what we meant to say in the previous manuscript. The point we mentioned is a subtle point regarding the implementation. In the conventional STDP protocol, pairing of a postsynaptic spike with a preceding presynaptic spike results in LTP while pairing with a succeeding one in LTD. In contrast, Masquelier et al only employed one of the plasticity effects arising from a pre-post pair with a shorter time difference. This implies that the postsynaptic neuron waits to decide between LTP and LTD until a future presynaptic spike arrives. We previously thought this rule is unrealistic. However, when the effect of a pre-post pairing takes place in synaptic plasticity is not a trivial issue (for instance, in the presence of dopamine). Therefore, we have eliminated the sentence.

- line 190 - 192, grammar/wording/logic

Reply – We have rewritten the sentence (line 213-214).

- line 264/265 ... and somatic output.

BETTER

... and somatic output in the presence of neuronal adaptation.

Reply – We have added the suggested phrase to the end of sentence (line 297).

- line 302

inhibitory circuits or intrinsic excitability.

ALSO MENTION

... or neuronal adaptation

Reply – We have mentioned “neuronal adaptation” and cited some references suggested by the reviewer (line 335-336).

- line 368/369

The reader may be confused at this stage by the superscript 'dend' on the distribution. You are talking at this stage only about a somatic distribution, and then this somatic distribution is rewritten as a normalized distribution. I suggest to drop the superscript 'dend' and instead note the (normalized) distribution as ϕ -hat (with a ^ for hat).

Reply – We agree with the reviewer that the previous superscript was misleading. We have changed the notation according to the suggestion by the reviewer (line 410).

- line 386-387 TURN SENTENCE AROUND, BECAUSE THE LOGIC IS NOT CLEAR

OLD:

We assumed that an attenuated version nu^ of dendritic potential well describes the somatic membrane potential with the degree of attenuation $alpha$ Explicitly representing ...*

NEW PROPOSITION:

In a stationary state, the somatic membrane potential u of a two-compartment model can be written as an attenuated version nu^ of the dendritic membrane potential with an attenuation factor $alpha = \dots$ (cite 21). Even though we work with time-dependent stimuli, we compare in our model at each point in time the attenuated dendritic membrane potential nu^* with the somatic membrane potential u . The comparison is not down directly on the level of the membrane potential but on the level of the two Poissonian spike distributions with rates $\phi_i^{som}(u)$ and ϕ_i^{hat} ($nustar$), respectively, that would be generated if both soma and dendrite were able to emit spikes independently. Explicitly representing ...*

Reply – Following the reviewer’s suggestion, we have rewritten the sentence at ll. 430-439. We agree that these modifications improve the mathematical clarity of our neuron model.

line 396

We search for ...

NEW PROPOSTION ...

Similar to reference (21), we search for ...

Reply – We have modified the sentence as suggested (line 453).

line 435

cite both Foldiak and Vogels.

Reply – We have cited these references below eq. (19).

line 473 The algorithm should be made publicly available on some web page or repository at the moment when the paper is accepted.

Reply – A github address has been added to the manuscript. Once the paper is accepted, we will open the github code to public. Please note currently the code is private.

References:

Adaptation and coding:

1. AL Fairhall, GD Lewen, W Bialek, RRR van Steveninck. *Efficiency and ambiguity in an adaptive neural code. Nature 412 (6849), 787*
2. M Maravall, RS Petersen, AL Fairhall, E Arabzadeh, ME Diamond. *Shifts in coding properties and maintenance of information transmission during adaptation in barrel cortex. PLoS biology 5 (2), e19*
3. S. Mensi, O. Hagens, W. Gerstner, and C. Pozzorini (2016) *Enhanced Sensitivity to Rapid Input Fluctuations by Nonlinear Threshold Dynamics in Neocortical Pyramidal Neurons. PLoS Comput Biol 12: e1004761. doi:10.1371/journal.pcbi.1004761*
4. C. Pozzorini, R. Naud, S. Mensi, and W. Gerstner (2013) *Temporal whitening by power-law adaptation in neocortical neurons. Nature Neuroscience 16:942 - 948*

LINK OF ICA and SFA and STDP

5. Clopath C, Longtin A, Gerstner W. *An online Hebbian learning rule that performs Independent component analysis. NIPS'07 Proceedings of the 20th International Conference on Neural Information Processing Systems, Pages 321-328, 2007*
6. H Sprekeler, T Zito, L Wiskott. *An extension of slow feature analysis for nonlinear blind source separation (2014) The Journal of Machine Learning Research 15 (1), 921-947*
7. H Sprekeler, C Michaelis, L Wiskott. *Slowness: An objective for spike timing-dependent plasticity? (2007) PLoS Computational Biology 3 (6), e112*
8. C Savin, P Joshi, J Triesch. *Independent component analysis in spiking neurons (2010) PLoS computational biology 6 (4), e1000757*

INHIBITORY PLASTICITY

10. TP Vogels, H Sprekeler, F Zenke, C Clopath, W Gerstner
Inhibitory Plasticity Balances Excitation and Inhibition in Sensory Pathways and Memory

Networks. Science 334 (6062), 1569-1573 (2011)

11. P. Földiák (1990) Forming sparse representations by local anti-Hebbian learning

Biol. Cybern. 64, 165-170 (1990)

I believe that after implementing these changes (and ideally with some additional comparisons) the manuscript is going to be a very nice paper for publication in Nature Communications that will find a broad readership

SIGNED

Wulfram Gerstner

Reply – We have cited the references recommended above in the revised manuscript together with a few additional references of our choice. We thank Prof. Gerstner for his careful review and valuable comments.

Reviewers' Comments:

Reviewer #2:

None